# *Xist* RNA binds select autosomal genes and depends on Repeat B to regulate their expression

**Shengze Yao[1,2], Yesu Jeon[1,2], Barry Kesner[1,2], Jeannie T Lee[1,2]***

[1]Department of Molecular Biology, Massachusetts General Hospital, Boston, United States; [2]Department of Genetics, The Blavatnik Institute, Harvard Medical School, Boston, United States

## eLife Assessment

This **valuable** study addresses the potential roles of the master regulator of X chromosome inactivation, the Xist long non-coding RNA, in the regulation of autosomal genes. Using data from mouse cells, the authors propose that Xist can coat specific autosomal promoters, which in turn leads to the attenuation of their transcriptional activity. The evidence from individual genes is interesting, and the model aligns with recently published results from humans. However, despite some improvements during revision, the data and statistical analyses in the current study are not yet strong enough to allow for conclusive inferences, leaving the evidence for mouse cells behaving like human cells **incomplete**. The topic of the work is of broad interest, in particular to colleagues studying gene regulation and noncoding RNAs.

**\*For correspondence:**
lee@molbio.mgh.harvard.edu

**Abstract** *Xist,* a pivotal player in X chromosome inactivation (XCI), has long been perceived as a cis-acting long noncoding RNA that binds exclusively to the inactive X chromosome (Xi). However, *Xist*'s ability to diffuse under select circumstances has also been documented, leading us to suspect that *Xist* RNA may have targets and functions beyond the Xi. Here, using female mouse embryonic stem cells (ES) and mouse embryonic fibroblasts (MEF) as models, we demonstrate that *Xist* RNA indeed can localize beyond the Xi. However, its binding is limited to ~100 genes in cells undergoing XCI (ES cells) and in post-XCI cells (MEFs). The target genes are diverse in function but are unified by their active chromatin status. *Xist* binds discretely to promoters of target genes in neighborhoods relatively depleted for Polycomb marks, contrasting with the broad, Polycomb-enriched domains reported for human *XIST* RNA. We find that *Xist* binding is associated with down-modulation of autosomal gene expression. However, unlike on the Xi, *Xist* binding does not lead to full silencing and also does not spread beyond the target gene. Over-expressing *Xist* in transgenic ES cells similarly leads to autosomal gene suppression, while deleting *Xist*'s Repeat B motif reduces autosomal binding and perturbs autosomal down-regulation. Furthermore, treating female ES cells with the *Xist* inhibitor, X1, leads to loss of autosomal suppression. Altogether, our findings reveal that *Xist* targets ~100 genes beyond the Xi, identify Repeat B as a crucial domain for its in-trans function in mice, and indicate that autosomal targeting can be disrupted by a small molecule inhibitor.

## Introduction

The unequal sex chromosome composition between male (XY) and female (XX) placental mammals necessitates dosage compensation of the X chromosome (*Payer and Lee, 2008*; *Disteche, 2016*). XCI specifically evolved to ensure equal X-linked gene dosage between the sexes. During early

embryogenesis, one of the two X chromosomes in female cells is randomly silenced, leading to a mosaic of cells in which X-linked genes can be expressed from either the maternal or paternal X chromosome (*Lyon, 1961*; *Deng et al., 2014*; *Lee, 2011*). The long non-coding RNA *Xist* is an essential regulator of the XCI process (*Brown et al., 1991*; *Penny et al., 1996*). At the onset of XCI, *Xist* is expressed exclusively from the future inactive X chromosome (Xi) and then selectively spreads in cis along the chromosome to initiate the formation of heterochromatin via association with chromatin-modifying complexes and alterations in 3D chromosome structure (reviewed in *Disteche, 2016*; *Jégu et al., 2017*; *Balaton et al., 2018*; *Martitz and Schulz, 2024*; *Sahakyan et al., 2018*; *Dixon-McDougall and Brown, 2021*). From early RNA fluorescence in situ hybridization (FISH) experiments, it was shown at a cytological level that *Xist* binds only to the X-chromosome which transcribes the RNA (*Brown et al., 1991*; *Clemson et al., 1996*). In *Xist* transgenesis studies, the RNA is also observed to localize exclusively in cis to the transgene, even on autosomes (*Lee et al., 1996*; *Lee et al., 1999*; *Wutz et al., 2002*; *Kohlmaier et al., 2004*; *Kelsey et al., 2015*; *Jiang et al., 2013*; *Minks and Brown, 2009*). In more recent years, technical advances in epigenomic mapping of RNA confirmed the early cytological data and provided a map of *Xist* binding sites in cis at kilobase resolution (*Simon et al., 2013*; *Engreitz et al., 2013*). Furthermore, genetic analysis of the X-inactivation center revealed a Xi-specific nucleation site for the initial binding of *Xist* that then enabled the RNA to spread exclusively in cis (*Jeon and Lee, 2011*). Altogether, these studies solidified the view that *Xist* RNA is a cis-acting RNA.

However, studies have long documented the potential for *Xist* to spread beyond the X-chromosome under various non-physiological conditions. When *Xist* is overexpressed, the RNA can diffuse to bind neighboring chromosomes (*Lee et al., 1999*; *Jeon and Lee, 2011*; *Jachowicz et al., 2022*). Furthermore, when *Xist*'s nucleation site is mutated, *Xist* will diffuse and bind other chromosomes in trans (*Jeon and Lee, 2011*). A recent study has also pointed to broad *XIST* binding patterns outside of the Xi in human naïve stem cells and suggested autosomal targets (*Dror et al., 2024*). These findings prompt interesting questions about the binding dynamics and properties of *Xist* on autosomes, its potential to spread locally in autosomal domains, and its impact on gene regulation and cellular physiology. To investigate, here we examine the capacity for *Xist* to spread beyond traditional boundaries. We use mouse embryonic stem cells (mESC) and MEF in order to capture both the establishment and maintenance phases of XCI. Intriguingly, we identify about 100 binding sites on autosomes in cells undergoing XCI as well as post-XCI cells. We demonstrate discrete binding sites, rather than broad binding domains. Transcriptomic analysis reveals a selective downregulation, but not silencing, of associated genes. We also probe a requirement for *Xist*'s Repeat B (RepB) motif and the ability to perturb autosomal effects by treating cells with a small molecule inhibitor of *Xist* RNA.

## Results

### Epigenomic mapping of *Xist* binding reveals autosomal targets in mouse cells

To explore the possibility of *Xist* extending its binding to autosomal sites beyond the X chromosome, we conducted CHART (Capture Hybridization of Associated RNA Targets) *Simon et al., 2013*; *Simon, 2013* in female ES cells. To capture *Xist* RNA, we used oligoprobes antisense to *Xist* and then pulled down interacting chromatin for deep sequencing of the associated DNA. To control for any direct hybridization to DNA that could occur independently of *Xist* RNA, we conducted CHART in parallel using 'sense' probes that should not hybridize to *Xist*. This helps us distinguish signals genuinely due to *Xist* RNA from those that could arise from non-specific probe binding to DNA. We note that some of the sense probes could hybridize to *Tsix* RNA, but *Tsix* expression is normally down-regulated by day 4 of ES differentiation (*Lee and Lu, 1999*). We also performed CHART using male ES cells, which do not upregulate *Xist* expression, to define the background level of binding. Notably, male ES cells express *Tsix* RNA at day 0 (*Lee and Lu, 1999*) and, therefore, also assist in excluding non-*Xist* RNA binding during analysis. Together the male and sense controls allowed us to account for non-specific interactions and background noise. In addition to these controls, we performed two biological replicates and analyzed only overlapping signals between the two replicates to ensure the reliability and reproducibility of our results. The Pearson and Spearman correlation analysis showed good reproducibility between replicates (*Figure 1—figure supplement 1A*). We also normalized to input DNA to account for differences in chromatin preparation and sequencing depth.

*Xist* RNA is transcriptionally upregulated during female ES cell differentiation when XCI is induced. To profile the dynamics during cell differentiation and XCI, we examined wild-type (WT) female ES cells in the undifferentiated state (day 0) and at three differentiated stages (day 4, 7, and 14). Antisense oligonucleotides targeting *Xist* efficiently pulled down *Xist* RNA and associated chromatin targets, in excess of the 'background' observed with the sense probe, and male control (*Figure 1—figure supplement 1B*). Principal component analysis (PCA) of CHART-seq data from day 4 and *Xist* coverage on X-linked genes revealed that *Xist* CHART signals in female cells were distinct from those of sense probe and male controls (*Figure 1—figure supplement 1C*). This PCA separation indicates a clear difference in *Xist* binding in females as compared to controls. As expected, the *Xist* CHART signal in WT female ES cells displayed strong binding at the *Xist* locus and genes subject to XCI, such as *Cdkl5* and *Mecp2*, while showing significantly weaker binding at the escapee gene, *Kdm6a*. In contrast, CHART experiments using sense probes and male ES cells, which lack *Xist* expression, showed minimal binding to these genes (*Figure 1—figure supplement 2A–D*, *Figure 1—figure supplement 3*). Because *Xist* spreads in cis along the Xi to cover much of the chromosome, we first examined total *Xist* coverage (normalized to input), rather than calling peaks (*Pinter et al., 2012*). Day 0 ES cells showed no enrichment for *Xist* binding, consistent with *Xist* being in the uninduced state (*Figure 1A*). Upon cell differentiation, enriched binding of *Xist* was consistently observed across all time points (*Figure 1A*). Greatest enrichment on the X chromosome was observed on day 7 (*Figure 1B*), in agreement with de novo XCI when *Xist* is needed at highest concentration (*Simon et al., 2013*; *Sunwoo et al., 2015*).

Intriguingly, on day 7, only ~10% of the reads mapped to the X chromosome (*Figure 1—figure supplement 1B*), prompting the question of whether *Xist* might have non-X-linked targets as well. Indeed, although the X chromosome was most enriched for *Xist*, we noticed that autosomes also showed highly reproducible peaks, beginning at day 4 and increasing across the differentiation time points (*Figure 1A*). In contrast to broad binding pattern covering the entire X chromosome (*Figure 1—figure supplement 2E*), *Xist* binding patterns on autosomes trended towards sharp peaks (*Figure 1C*, *Figure 1—figure supplement 2F*). To determine if the peaks were statistically significant, we called *Xist* peaks using MACS2 on the *Xist* CHART data. We restricted the analysis to autosomal reads to increase the sensitivity of the analysis. We performed peak calling separately for each CHART biological replicate. Irreproducible Discovery Rate (IDR) analysis indicated a strong correlation between the two replicates (*Figure 1—figure supplement 4A*). We used *bedtools intersect* to identify significant peaks that overlapped between the two replicates and only used overlapping peaks in subsequent analyses. Overall, *Xist* coverages in significant peaks of female cells were substantially greater than in the negative controls (*Figure 1E and F*).

Intriguingly, we found hundreds of significant peaks across autosomes between days 4–14 of differentiation (*Figure 1D*, *Figure 1—figure supplement 4B*, *Supplementary files 1-6*). Notably, *Xist* binding sites on autosomes were disproportionately located in promoter regions (*Figure 1A and G*), hinting at a potential role for *Xist* in autosomal gene regulation. We conclude that, in addition to the Xi of differentiating ES cells, mouse *Xist* RNA selectively binds ~100 autosomal targets, preferentially at promoter regions. This promoter-dominant profile contrasts with *XIST* patterns identified in human cells, which tend to demonstrate broad regions of coverage over genes (*Dror et al., 2024*). Although the Gene Ontology (GO) analysis of these *Xist* autosomal target genes did not yield statistically significant enrichment for specific biological processes or pathways, a closer examination reveals that these genes have diverse functions. They are involved in processes such as cancer development (*Bcl7b*), POLII transcription regulation (*Med16*), amino acid transport (*Slc36a4*), RNA binding (*Rbm14* and *Stau2*), among other processes. Together with previously published work in human cells (*Dror et al., 2024*), our findings suggest that *Xist* may have a broad regulatory impact on a variety of autosomal genes, extending its influence beyond X-chromosome inactivation.

## Deleting RepB causes a loss of binding to autosomal target genes

Prior work showed that *Xist*'s Repeat B (RepB) element is essential for Polycomb recruitment (*Almeida et al., 2017*; *Pintacuda et al., 2017*; *Colognori et al., 2019*; *Colognori et al., 2020*). RepB is also required for proper spreading and localization of *Xist* RNA to the Xi: Without RepB, the *Xist* RNA cloud was observed to adopt a dispersed appearance consistent with diffusion of the RNA away from the Xi (*Colognori et al., 2019*). Given the partial loss of attachment to the Xi, here we asked if the

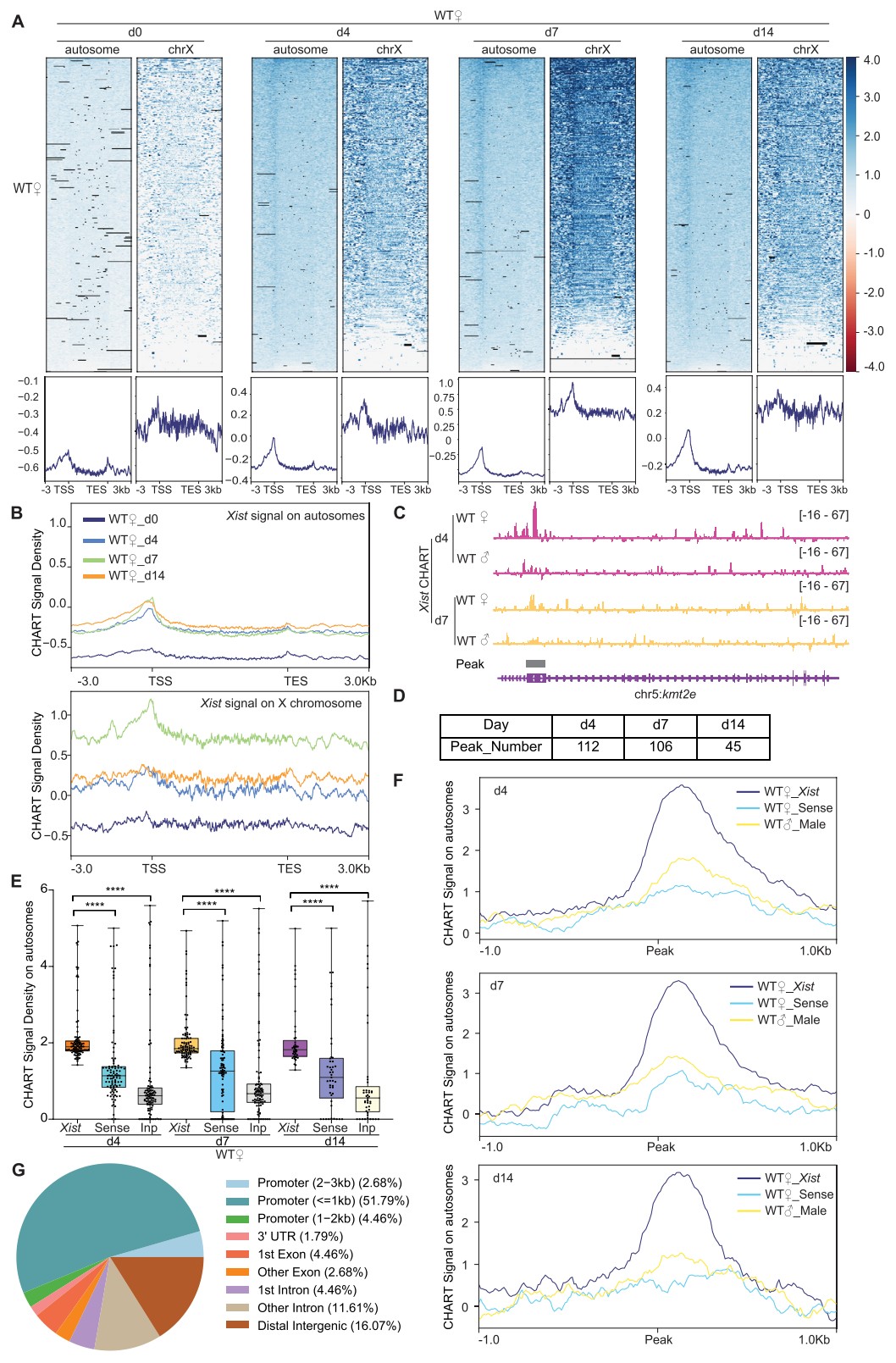

**Figure 1.** Capture hybridization analysis of RNA targets (CHART)-seq reveals ~100 discrete binding sites in autosomal genes. (**A**) Top: Heatmaps of *Xist* coverage on autosomes and chromosome X genes in wild-type (WT) female embryonic stem (ES) cells at day 0, day 4, day 7, and day 14. Bottom: Average profiles shown as metagene maps of the data in the top panel. TSS, transcription start site. TES, transcription end site. (**B**) Average profile of

*Figure 1 continued*

*Xist* coverage on autosomes and X chromosome genes in WT female ES cells at day 0, day 4, day 7, and day 14. (**C**) Representative site-specific binding of *Xist* on autosome locus (*Kmt2e*) in WT female ES cells at day 4 and day 7, WT male ES cells are used as control. (**D**) Number of *Xist* peaks determined by MACS2 peak calling on autosomes in WT female ES cells at day 4, day 7, and day 14. (**E**) *Xist* CHART signal coverage on *Xist*-autosomal peak region (top 100 peaks) in WT female ES cells at day 4, day 7, and day 14. Sense and input are used as control. P-values are determined using the Wilcoxon rank sum test. (**F**) Average profile of CHART (subtracted input) signal on *Xist*-autosomal peak region (top 100 peaks) in WT female and male ES cells at day 4, day 7, and day 14. Sense and WT male ES are used as control. (**G**) Feature annotation of *Xist* binding loci on autosomes by ChIPseeker in WT female ES cells at day 4.

The online version of this article includes the following figure supplement(s) for figure 1:

**Figure supplement 1.** *Xist* binds autosomal genes in trans.

**Figure supplement 2.** *Xist* binds to the X chromosome during female embryonic stem (ES) cell differentiation.

**Figure supplement 3.** *Xist* binds X chromosome inactivation (XCI) genes during differentiation.

**Figure supplement 4.** *Xist* binds select autosomal genes in trans.

*Xist* diffusion could lead to enhanced binding to autosomal targets. We conducted CHART experiments in RepB deletion (ΔRepB) female ES cells at various differentiation stages and quantitated the degree of X-linked versus autosomal binding (***Figure 2***). Although *Xist* retained partial binding to the X-chromosome, there was a significant decrease in localization along X-linked genes along all differentiation days, especially at day 7 (***Figure 2A***), consistent with RepB being essential for attachment of *Xist* to the Xi (***Colognori et al., 2019***). Intriguingly, however, the loss of Xi binding was not accompanied by any significant increase in *Xist* coverage at the same autosomal targets, at any differentiation day (***Figure 2A***). In fact, there appeared to be decreased binding to the ~100 autosomal genes (***Figure 2E,F***, ***Figure 1—figure supplement 4C-D***). However, there was not a change up- or down-wards in the number of autosomal targets (***Figure 2B-C***, ***Figure 1—figure supplement 4B***, ***Figure 1—figure supplement 4E***, ***Supplementary files 1-6***). A similar number of *Xist* peaks across autosomes in ΔRepB cells was observed and the autosomal targets remained similar. Moreover, in the absence of RepB, *Xist* retained a preference to bind promoter regions (***Figure 2D***). Thus, RepB is required not only for *Xist* to localize to the X-chromosome but also for its localization to the ~100 autosomal genes.

## *Xist* RNA down-regulates autosomal genes in a RepB-dependent manner

The promoter-specific binding pattern of *Xist* on autosomal targets contrasts sharply with the broad binding pattern of *Xist* on the Xi (***Figure 1A***) and with broad regions identified for human autosomal genes (***Dror et al., 2024***). Given the preference for autosomal promoters, we investigated whether *Xist* modulates expression of the associated autosomal genes. To address this question, we conducted transcriptome sequencing on WT female ES cells across day 0, 4, 7, and 14, and compared the profile to those of ΔRepB female ES cells. The RNA-seq data showed that *Xist* expression increased during the establishment phase of XCI in differentiating ES cell, reaching its peak at day 7 (***Figure 2—figure supplement 1A***), consistent with super-resolution data indicating that *Xist* is present at ~300 copies/cell during de novo XCI relative to the ~100 copies/cell during the maintenance phase (***Sunwoo et al., 2015***). *Xist* expression followed a similar dynamic in ΔRepB female cells (***Figure 2—figure supplement 1A***). However, deletion of *Xist*'s RepB resulted in increased X-linked genes expression in differentiating female ES cells (***Figure 3D***, ***Figure 2—figure supplement 1B-C***), consistent with a previous report (***Colognori et al., 2019***). To analyze the characteristics of genes bound by *Xist* on autosomes, we categorized all refSeq genes based on their expression levels at each differentiation day. The genes were divided into five quintiles (Q1 silent, Q2 low, Q3 moderate, Q4 high, Q5 highest). Consistent with *Xist*'s behavior on the X-chromosome (***Simon et al., 2013***; ***Engreitz et al., 2013***). *Xist* also favored binding to actively expressed genes on autosomes (***Figure 3A***). To examine gene expression levels as a function of distance from the *Xist* binding site, we divided neighboring genes into 10- to 100 kb bins and observed that genes at <10 kb showed highest *Xist* levels (***Figure 3B***, ***Figure 3—figure supplement 1A***). These findings support *Xist*'s favoring actively expressed genes.

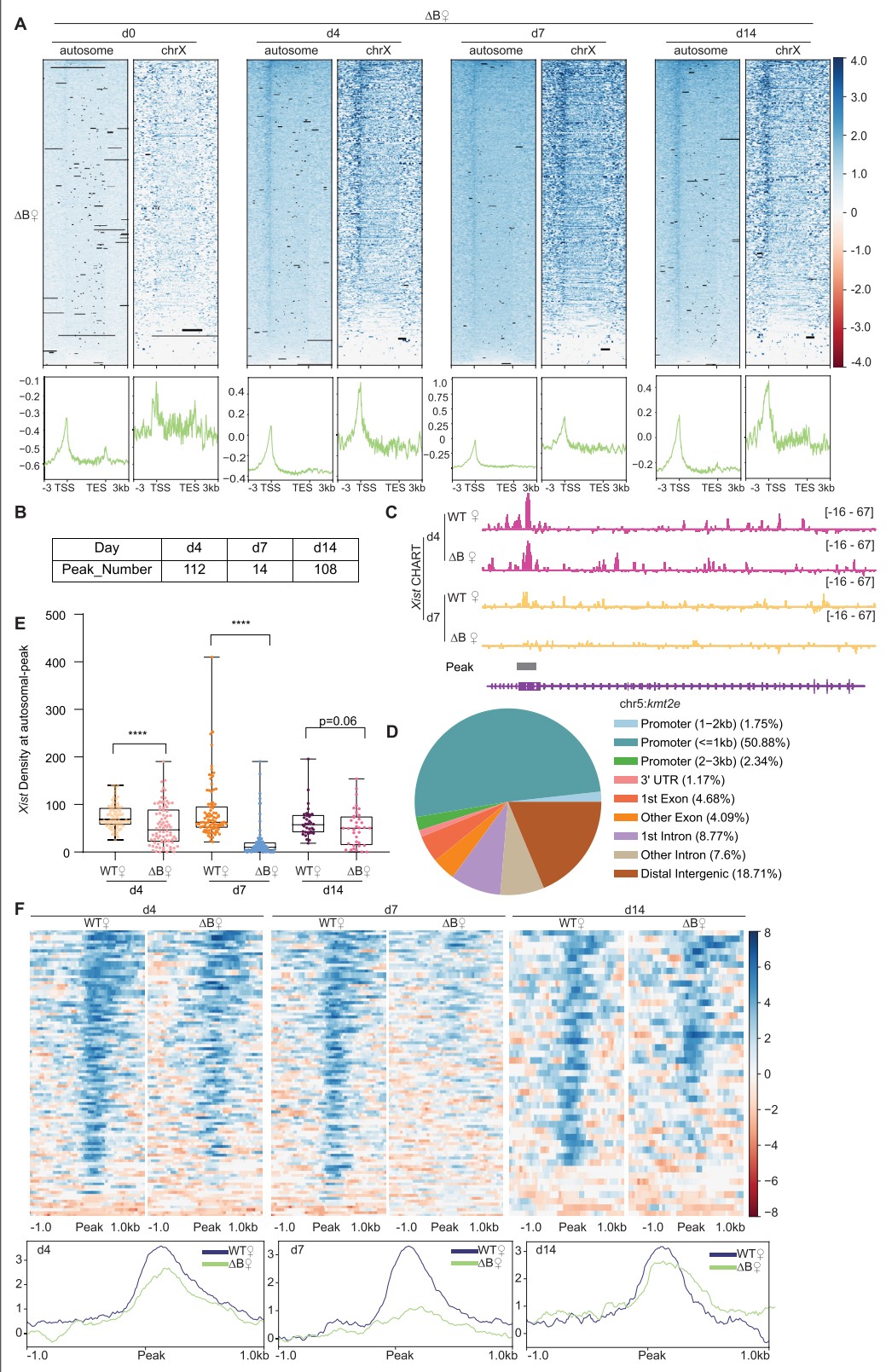

**Figure 2.** *Xist*'s Repeat B is required for proper binding of *Xist* to autosomal genes. (**A**) Top: Heatmaps of *Xist* coverage on autosomes and chromosome X genes in wild-type (WT) female embryonic stem (ES) cells at day 0, day 4, day 7, and day 14. Bottom: Average profiles shown as metagene maps of the data in the top panel. (**B**) Number of *Xist* peaks determined by MACS2 peak calling on autosomes in ΔRepB female ES cells at day 4, day 7,

*Figure 2 continued on next page*

*Figure 2 continued*

and day 14. (**C**) Representative site-specific binding of *Xist* on autosome locus (*Kmt2e*) in WT and ΔRepB female ES cells at day 4 and day 7. (**D**) Feature annotation of *Xist* binding loci on autosomes by ChIPseeker in ΔRepB female ES cells at day 4. (**E**) *Xist* capture hybridization analysis of RNA targets (CHART) signal coverage on *Xist*-autosomal peak region in WT and ΔRepB female ES cells at day 4, day 7, and day 14. P-values are determined using the Wilcoxon rank sum test. (**F**) Top: Heatmaps of *Xist* coverage on *Xist*-autosomal peak region (top 100 peaks) in WT and ΔRepB female ES cells at day 4, day 7, and day 14. Bottom: Average profiles shown as metagene maps of the data in the top panel.

The online version of this article includes the following figure supplement(s) for figure 2:

**Figure supplement 1.** *Xist* binds X-linked genes is *Xist*'s RepB dependent.

We then examined Polycomb marks in relation to the autosomal *Xist* domains. In agreement with the *Xist*'s predilection for active genes, we observed lower coverage of Polycomb marks associated with *Xist* RNA (*Dixon-McDougall and Brown, 2021*; *Kohlmaier et al., 2004*; *Almeida et al., 2017*; *Pinta-cuda et al., 2017*; *Colognori et al., 2019*; *Colognori et al., 2020*; *Zhao et al., 2008*), H3K27me3, and H2AK119ub (associated with PRC2 and PRC1, respectively), as shown by lower coverages within the 10 kb versus 50 kb neighborhoods (*Figure 3C*, *Figure 3—figure supplement 1C*). These findings also contrasted with observations in human cells, where PRC2/H3K27me3-enriched regions were favored (*Dror et al., 2024*). Our findings indicate that WT *Xist* transcripts favor binding to a select set of actively transcribed autosomal genes and these genes are not marked by Polycomb in mouse cells.

Given the association with active autosomal genes, we next asked whether *Xist*'s binding modulates their expression. We performed RNA-seq analysis in two biological replicates in the ES differentiation time series. First, we profiled gene expression in female ES cells relative to male ES cells, reasoning that the lack of *Xist* RNA in male ES cells might lead to a difference in female versus male expression of autosomal target genes. Indeed, among the ~100 target genes, there was significantly greater expression in male cells (*Figure 3E–G*), suggesting that *Xist* might also negatively regulate the select genes on autosomes. To test this idea, we took advantage of the ΔRepB mutation, as RepB is required for proper silencing of Xi genes (*Colognori et al., 2019*). Transcriptomic analysis demonstrated that perturbing RepB resulted in loss of repression of autosomal targets. A detailed examination focusing on the top 100 peaks of *Xist*-bound genes on autosomes unveiled a down-modulation especially evident on days 4 and 7 of ES cell differentiation (*Figure 3G*, *Figure 3—figure supplement 2A-D*). The *Xist*-bound autosomal genes on day 14 of ES cell differentiation like *Ces1l* or non-targets of *Xist* on autosomes such as *Kmt2c* did not demonstrate significant changes in gene expression (*Figure 3—figure supplement 2E–F*). We note, however, that unlike on the Xi, autosomal binding of *Xist* did not lead to full silencing of the target gene. Rather, the effect was a partial repression. In ΔRepB cells, reduced autosomal binding resulted in a blunting of this repression (*Figure 3F-G*, *Figure 3—figure supplement 2A-D*). On the other hand, at day 0 when *Xist* had yet to be upregulated, ΔRepB female and male ES cells showed no significant expression changes at *Xist*-autosomal targets (*Figure 3H*). To confirm that the observed changes in autosomal genes are due to *Xist* binding, we performed a similar analysis on 100 randomly selected autosomal genes that did not have *Xist* binding ('on-targets'). The non-targets did not show a preference for specific autosomal gene region (*Figure 3—figure supplement 3A–C*) and showed no significant difference in Polycom enrichment (*Figure 3—figure supplement 3D*) or gene expression (*Figure 3—figure supplement 3E*). These results demonstrate that autosomal binding of *Xist* RNA directly leads to a suppression of gene expression in a RepB-dependent manner.

## Effects of *Xist* overexpression and X1 drug treatment on autosomal genes

To seek further evidence for autosomal effects of *Xist* binding, we conducted two orthogonal lines of experimentation. First, we asked the reciprocal question and determined whether ectopically overexpressing *Xist* leads to increased autosomal repression. We examined two previously published transgenic *Xist* cell lines, Tg_1 and Tg_2 (*Loda et al., 2017*), to test whether — *in the hands of other investigators* — cell lines expressing *Xist* also demonstrated autosomal targeting (*Figure 4A*). These two transgenic (Tg) female cell lines respond to doxycycline to induce *Xist* expression on autosomes (specifically chromosome 12, with two different transgene insertion sites). Transcriptomic analysis of

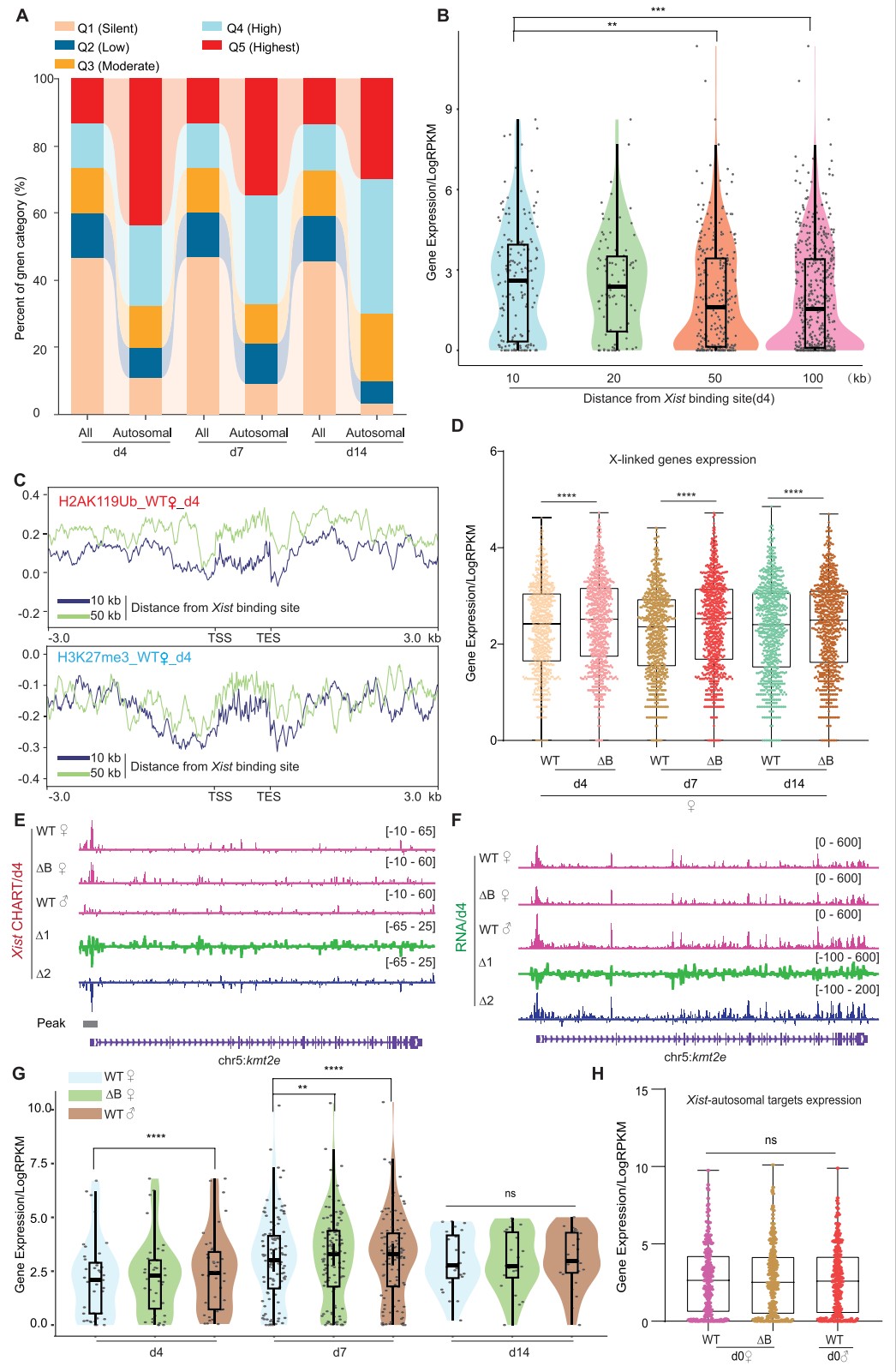

**Figure 3.** Autosomal target genes are actively transcribed. (**A**) *Xist* preferentially targets actively expressed genes. The proportion of overall genes (labeled as 'All') and *Xist* targets on autosomes expression levels (labeled as 'Autosomal') in different expression categories. Genes are classified into distinct categories (**Q1–Q5**) based on Reads Per Kilobase Million (RPKM) values, representing various levels of expression. Category Q1 includes

*Figure 3 continued on next page*

*Figure 3 continued*

non-expressed genes (RPKM = 0), while categories Q2, Q3, Q4, and Q5 represent 25%, 50%, 75%, and 100% expression, respectively. (**B**) Genes bound by *Xist* exhibit elevated expression levels compared to surrounding regions. The analysis of gene expression within the 10, 20, 50, and 100-kilobase binding regions of *Xist* is conducted in wild-type (WT) female embryonic stem (ES) cells at day 4. p-values are determined using the Wilcoxon rank sum test. (**C**) The H2AK119ub and H3K27me3 (ChIP-Seq) average profile of genes within the 10 and 50-kilobase binding regions of *Xist* in WT female cells at day 4. (**D**) Evaluation of gene expression levels for X chromosome inactivation (XCI) genes in WT and ΔRepB female ES cells at day 4, day 7, and day 14. P-values are determined using the Wilcoxon rank sum test. (**E–F**) Representative capture hybridization analysis of RNA targets (CHART)-seq (**F**) and RNA-Seq (**G**) patterns of an autosomal gene bound by *Xist (Kmt2e)* at day 4. Change in coverage (Δ1 and Δ2) is shown below (Δ1 for ΔRepB♀ -WT♀, and Δ2 for WT♂ -WT♀). (**G**) Assessing gene expression levels of *Xist* targets on autosomes (10 kb within the peak region) in WT, ΔRepB female ES cells, and male ES cells at different time points. p-values are determined using the Wilcoxon rank sum test. (**H**) Gene expression levels for *Xist* targets on autosomes (identified in day 4 and day 7) in undifferentiated WT, ΔRepB, female and male ES cells (day 0) show no obvious changes. Two biological replicates were used. p-values are determined using the Wilcoxon rank sum test.

The online version of this article includes the following figure supplement(s) for figure 3:

**Figure supplement 1.** Genes bound by *Xist* exhibit higher expression levels and lower H3K27me3 and H2AK119ub binding levels.

**Figure supplement 2.** Example of *Xist* binding on autosomal genes and influence on gene expression.

**Figure supplement 3.** Genes not bound by *Xist* exhibit no changes in gene expression or differences in H3K27me3 and H2AK119ub signals.

these two Tg mouse ES cells upon neuronal differentiation showed a significant suppression of X-linked genes (e.g. *Med14*) in comparison to doxycycline-treated wildtype cells (***Figure 4B***, ***Figure 4—figure supplement 1A***). Autosomal *Xist* targets, as exemplified by *Bcl7b* and *Rbm14*, were also significantly suppressed beyond what was observed in WT ES cells (***Figure 4C***, ***Figure 4—figure supplement 1B***). In contrast, non-targets of *Xist* such as *Stau1* did not demonstrate significant changes in gene expression (***Figure 4E and F***). Looking across all autosomal target genes, we observed a significant decrease in mean expression in the *Xist* overexpressing cell lines (***Figure 4D***), further implicating *Xist* binding in direct regulation of autosomal gene targets. The fact that the autosomal changes were also observed in datasets generated by other investigators strengthen our conclusions.

In the second line of investigation, we deployed X1, a small molecule inhibitor of *Xist* that was previously shown to block the initiation of XCI in female ES cells through *Xist*'s Repeat A (RepA) motif (***Aguilar et al., 2022***). We reasoned that if the mechanism of autosomal gene suppression were similar to XCI, treating cells with X1 could also blunt autosomal gene suppression. Indeed, just as X-linked genes such as *Mecp2* failed to undergo silencing after 5 d of differentiation, transcriptomic analysis showed that autosomal gene targets also failed to be suppressed in the presence of X1 (***Figure 5A–C***). On average, there was a significant upregulation of autosomal targets on day 5 of differentiation and X1 treatment (***Figure 5E***). Again, non-targets of *Xist* such as *Cbx2* did not demonstrate significant changes in gene expression (***Figure 5D and F***). Analysis of ChIP-seq data for H3K27me3 indicated a reduction of the PRC2 mark within the *Xist* target gene and flanking regions (***Figure 5G and H***), consistent with X1 blocking RepA's recruitment of PRC2 (***Aguilar et al., 2022***). In contrast, *Xist* non-target genes show no significant changes in PRC2 signal (***Figure 5I***). These additional perturbation studies reinforce the notion that autosomal binding of *Xist* RNA directly suppresses autosomal targets. Thus, it is possible to disrupt autosomal *Xist* binding by administering a small molecule inhibitor of *Xist*.

## *Xist* RNA also binds to autosomal genes in post-XCI cells

Lastly, because the establishment and maintenance phases of XCI show differential sensitivity to loss of *Xist* (***Jacobson et al., 2022***; ***Brown and Willard, 1994***; ***Csankovszki et al., 1999***; ***Zhang et al., 2007***), we asked if autosomal repression exhibits a similar sensitivity. To address this, we turned to MEF, where XCI is well established. We first conducted CHART assays to determine whether *Xist* binds to the Xi and autosomes in a similar way in MEFs (***Figure 6—figure supplement 1A–B***). Because the autosomal signals displayed peak-like characteristics, we used MACS software to call significant peaks on autosomes to determine if there was an increase in promoter-specific binding of the autosomal

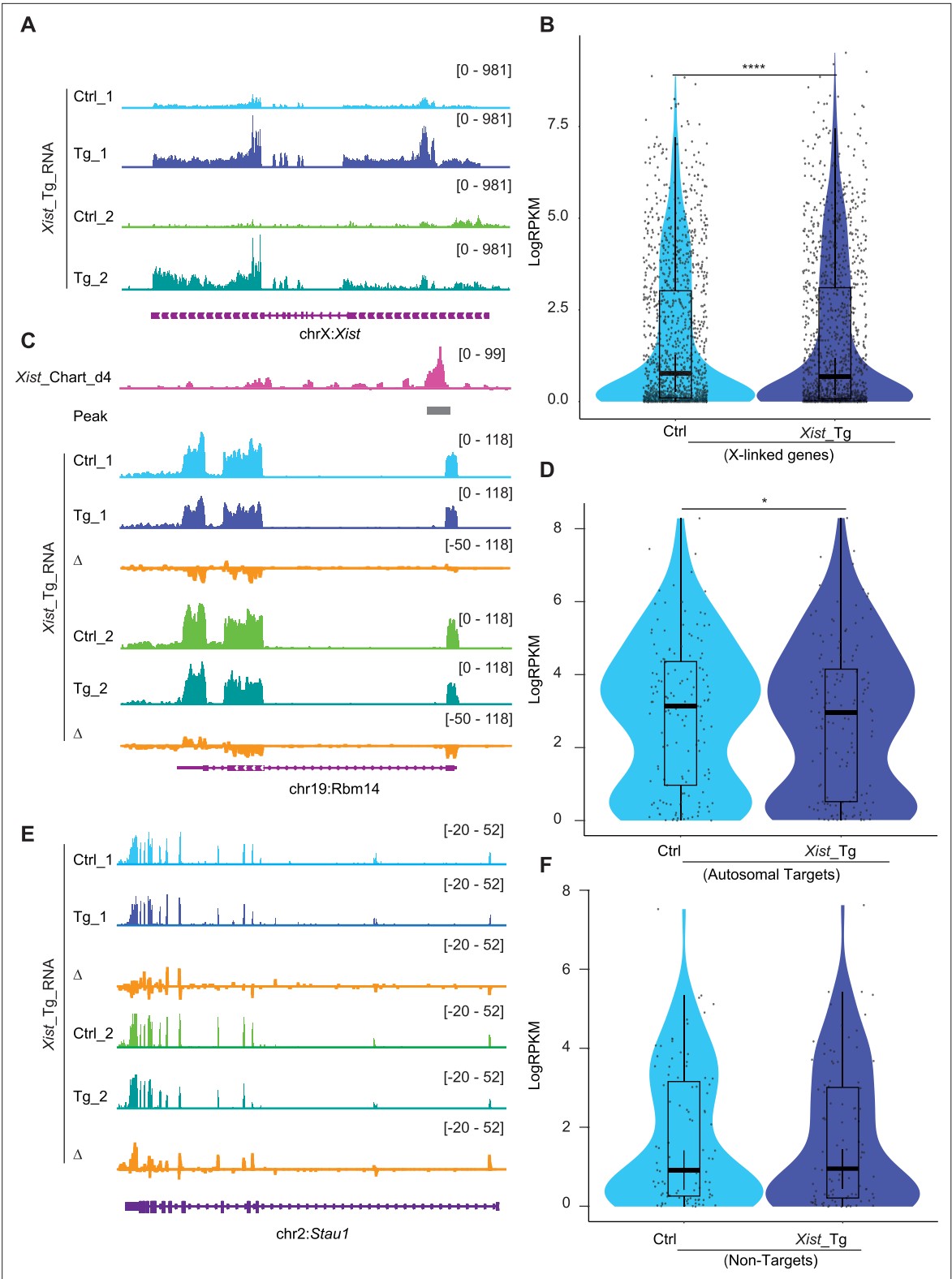

**Figure 4.** *Xist* binding suppresses, but does not silence, autosomal gene expression. (**A**) RNA-seq track shows the *Xist* expression level in ectopic doxycycline-responsive *Xist* overexpressed embryonic stem (ES) cells (129/Sv-Cast/Ei) of neuronal differentiation (Tg-1 and Tg-2 have different insertion sites), the control (Ctrl) was doxycycline-treated wildtype cell. (**B**) Analysis of gene expression levels for X chromosome inactivation (XCI) genes in *Xist* Tg ES cells (129/Sv-Cast/Ei) at neuronal differentiation. p-values are determined using the Wilcoxon rank sum test. (**C**) RNA-seq track illustrating expression alongside representative autosome genes with *Xist* binding (*Rbm14*) in *Xist* Tg ES cells (129/Sv-Cast/Ei) of neuronal differentiation. Change

*Figure 4 continued on next page*

*Figure 4 continued*

in coverage (Δ) is shown below (Tg - Ctrl). (**D**) Analysis of gene expression levels for *Xist* autosomal targets in *Xist* Tg ES cells (129/Sv-Cast/Ei) of neuronal differentiation. p-values are determined using the Wilcoxon rank sum test. (**E**) RNA-seq track illustrating expression level of autosome genes without *Xist* binding (*Stau1*) in *Xist* Tg ES cells (129/Sv-Cast/Ei) of neuronal differentiation. Change in coverage (Δ) is shown below (Tg - Ctrl). (**F**) Analysis of gene expression levels for *Xist* non-targets on autosomes in *Xist* Tg ES cells (129/Sv-Cast/Ei) of neuronal differentiation. p-values are determined using the Wilcoxon rank sum test.

The online version of this article includes the following figure supplement(s) for figure 4:

**Figure supplement 1.** *Xist* overexpression inhibits select autosomal genes and X-linked genes.

genes. We observed that, while *Xist* continued to favor targeting promoter regions in both WT and ΔRepB MEFs (*Figure 6A–C*), *Xist* peaks did not increase in size in ΔRepB MEFs; rather, the peaks decreased in size (*Figure 6A–C*) — a result similar to that in ΔRepB ES cells (*Figure 2*). Furthermore, there was a considerable decrease in the number of significant peaks in ΔRepB MEFs when compared to WT MEFs (*Figure 6A*). In WT MEFs, *Xist* clearly favored Xi binding, but *Xist* signals could also be visualized on autosomes (*Supplementary files 7-10*, *Figure 6E*), in agreement with the results in ES cells (*Figure 1*). In ΔRepB MEFs, there was a partial loss of Xi localization, aligning with previous findings (*Colognori et al., 2019*; *Figure 6D–F*). Consistent with these findings, RNA-seq analysis of ΔRepB MEFs did not reveal any significant change in expression of Xi genes (e.g. *Mecp2*) and auto-somal target genes (e.g. *Rbm14*) (*Figure 7A–D*).

We then asked if the retention of some *Xist* binding on the Xi could explain the lack of transcriptomic difference in ΔRepB MEFs. To investigate, we utilized a deletion of *Xist*'s Repeat E (ΔRepE), which was previously demonstrated to severely abrogate localization of *Xist* to the Xi (*Sunwoo et al., 2017*; *Ridings-Figueroa et al., 2017*). We reasoned that the severe loss of *Xist* binding might unmask a transcriptomic difference. As expected, we observed that *Xist* signals were somewhat more reduced on the Xi in ΔRepE MEFs compared to ΔRepB cells (*Figure 6E–F*). Despite this reduction, peak coverages in autosomal target genes did not increase in ΔRepE MEFs (*Figure 6E–F*). However, there was an overall decrease in the number of significant autosomal peaks in ΔRepE MEFs relative to WT cells (*Figure 6A*). Regardless, we observed no significant transcriptomic differences in ΔRepE MEFs relative to WT MEFs (*Figure 7A–E*). Additionally, further examination of RNA sequencing data from male and female MEF cells in two published studies (*Mizukami et al., 2019*; *Belužić et al., 2024*) corroborated that the expression levels of these autosomal *Xist* targets did not exhibit significant changes (*Figure 7F and G*). Altogether, the analysis in MEFs demonstrates that *Xist* continues to bind auto-somal genes in post-XCI somatic cells. However, autosomal binding of *Xist* in post-XCI cells does not overtly impact the expression of the associated autosomal genes. Nonetheless, we cannot exclude more subtle changes that do not meet the significance cut-off.

## Discussion

Together with a backdrop of studies on ability of *Xist* RNA to diffuse and bind chromatin in trans (*Lee et al., 1999*; *Jeon and Lee, 2011*; *Jachowicz et al., 2022*; *Dror et al., 2024*), the results of our current work challenge the conventional narrative that *Xist* operates exclusively in cis on the Xi. We find that autosomal *Xist* targets are not numerous, possibly limited to only ~100 in both plurip-otent stem cells and in somatic cells. On autosomes, *Xist* does not spread and instead covers only a narrow region corresponding to single genes. The genes tend to be active genes, with *Xist* specifically targeting the promoters of those genes. Genetic analysis coupled to transcriptomic analysis showed that *Xist* down-regulates the target autosomal genes without silencing them. This effect leads to clear sex difference — where female cells express the ~100 or so autosomal genes at a lower level than male cells in the mouse ES cell differentiation (*Figure 7H*). Thus, our findings redefine the scope of *Xist*'s functional repertoire and provide insights into the broader landscape of epigenetic regulation during cellular differentiation. *Xist* RNA, therefore, plays a more complex role than previously envis-aged, with several implications and caveats.

First, the observed consistent binding of *Xist* to both the X chromosome and autosomes in female ES cells prompted further exploration of the intricate dynamics of *Xist* during cellular differentia-tion. Our categorization of genes based on their expression levels revealed a compelling correlation between *Xist* binding on autosomes and the expression levels of associated genes. Notably, *Xist*

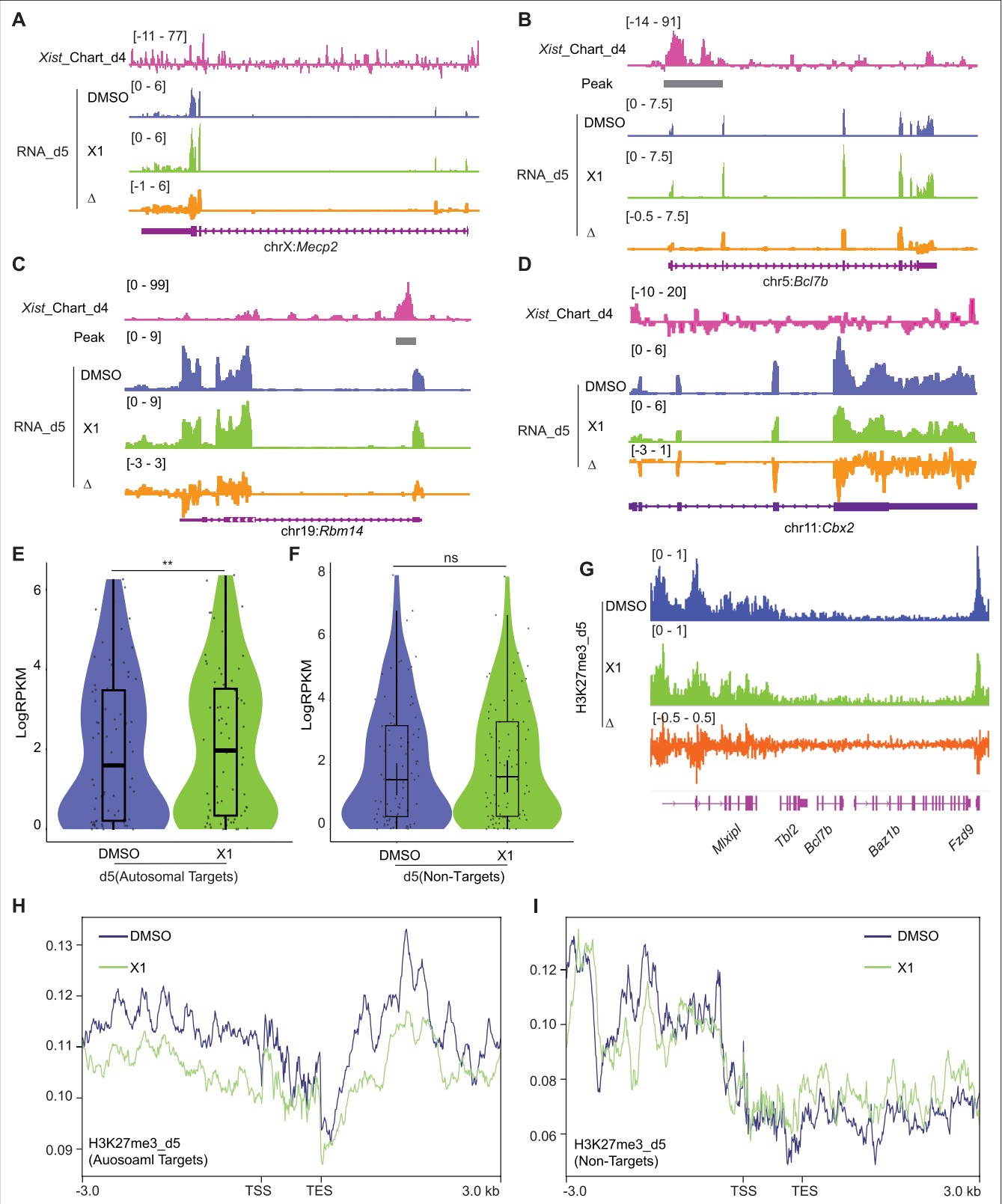

**Figure 5.** Treating cells with the X1 inhibitor of *Xist* RNA perturbs autosomal target genes. (**A–D**) RNA-seq track depicting *Mecp2* (**A**) (an X-linked gene), *Bcl7b* (**B**) and *Rbm14* (**C**) (*Xist*-regulated autosome target), and *Cbx2* (**D**) (autosomal gene without *Xist* binding) expression in DMSO (control) and *Xist* inhibitor (**X1**) treated differentiated wild-type (WT) female embryonic stem (ES) cells at day 5 of differentiation. Change in coverage (Δ) is shown below (X1 - DMSO). The *Xist* peak on autosomes is from capture hybridization analysis of RNA targets (CHART) at day 4. (**E**) Evaluation of gene expression

*Figure 5 continued on next page*

*Figure 5 continued*

levels for *Xist* autosomal targets in DMSO (control) and X1 treated WT female ES cells at day 5 of differentiation. The *Xist* peak on autosomes is from CHART at day 4. p-values are determined using the Wilcoxon rank sum test. (**F**) Evaluation of gene expression levels for *Xist* non-targets on autosomes in DMSO (control) and X1 treated WT female ES cells at day 5 of differentiation. p-values are determined using the Wilcoxon rank sum test. (**G**) ChIP-seq track for H3K27me3 on *Bcl7b* (an *Xist* autosomal target, within a 160 kb window) in DMSO (control) and X1 treated WT female ES cells at day 5 of differentiation. Change in coverage (Δ) is shown below (X1 - DMSO). (**H–I**) Profile plot displaying H3K27me3 levels for *Xist*-autosomal targets flanking genes (**H**) (within a 100 kb window) or *Xist* non-targets (**I**) in DMSO (control) and X1 treated WT female ES cells at day 5 of differentiation. The *Xist* autosomal target is from CHART at day 4.

exhibited a preference for binding to active gene regions, as evidenced by the significantly higher expression of genes within the 10 kb range of *Xist* binding regions. The parallel upregulation of X-linked and autosomal genes in ΔRepB and male ES cells suggests a regulatory role of *Xist* in gene expression beyond its canonical function on the X chromosome. Therefore, the precise temporal modulation of *Xist* binding merits further investigation to elucidate its regulatory significance and during distinct stages of cell differentiation or development, particularly in understanding its autosomal targets which could have implications for gene regulation and cellular function.

Second, a recent study using an alternative *Xist* pulldown method, RAP-seq, also revealed the capability of *Xist* to bind to autosomes in naive human pluripotent stem cells (naive hPSCs) and mediate gene regulation (***Dror et al., 2024***). While we found that, on autosomes, *Xist* does not spread and instead covers only a narrow region corresponding to single genes, the previous study demonstrated broader regions of binding (***Dror et al., 2024***). Furthermore, our CHART experiments demonstrate *Xist* binding to autosomes in post-XCI cells (e.g. MEF), whereas the previous study found no significant binding beyond the stem cell stage (***Dror et al., 2024***). Our analysis also indicates that *Xist*'s autosomal targets tend to be Polycomb-depleted relative to neighboring genes, whereas the prior human study observed a positive correlation between *XIST* RNA and PRC2 marks. These discrepancies may arise from differences in the *Xist* pulldown methods (CHART versus RAP) employed by the two studies or from inherent differences between mouse and human systems.

Third, we considered the possibility that the binding of *Xist* to autosomes could merely be a consequence of *Xist* diffusion following saturation of binding sites on the Xi, rather than any programmed event during development. We are inclined to reject this notion and propose that *Xist* binding to autosomes is specifically programmed. Upon deleting RepB, the binding of *Xist* to the X chromosome weakens, but concomitantly, its binding to autosomal targets also diminishes (***Figure 2***). This suggests that *Xist* binding to autosomes is contingent upon Repeat B and is deliberate, rather than due to random *Xist* diffusion alone (in which case we would have expected increased autosomal binding). Additionally, following treatment with the *Xist* inhibitor X1, an increase in the expression of autosomal targets is observed (***Figure 5***), implying that the regulation of autosomes by *Xist* is not merely an overflow effect of *Xist* saturating the X chromosome.

In summary, our study advances understanding of *Xist*-mediated epigenetic regulation by highlighting unexplored interactions with autosomes. The identified correlations between *Xist* binding and gene expression involvement pose intriguing questions regarding the regulatory mechanisms governing these processes. These insights contribute to the evolving paradigm of *Xist* biology, underscoring the need for continued exploration of the complexity of epigenetic control mechanisms. Future investigations will delve deeper into the functional consequences of *Xist* binding on autosomes and explore the potential downstream effects on cellular differentiation, development, and diseases associated with its dysregulation, such as cancer, immunity, and neuron development (***Dror et al., 2024***; ***Forsyth et al., 2024***; ***Pyfrom et al., 2021***; ***Yildirim et al., 2013***; ***Carrette et al., 2018***; ***Hajdarovic et al., 2022***). A comprehensive understanding of *Xist*'s influence beyond X chromosomes is crucial, particularly laying the groundwork for future exploration of *Xist* as a potential therapeutic target.

# Methods

## Key resources table

| Reagent type (species) or resource | Designation | Source or reference | Identifiers | Additional information |
|---|---|---|---|---|
| Cell line (*Mus musculus*) | female-wild-type MEF (*M. musculus*/M. castaneus F1 hybrid) | PMID:30827740 | EY.T4 | Maintained in Jeannie T Lee lab |
| Cell line (*Mus musculus*) | female-Xist Repeat B KO MEF (*M. musculus*/M. castaneus F1 hybrid) | PMID:30827740 | EY.T4-Xist RepB KO | Maintained in Jeannie T Lee lab |
| Cell line (*Mus musculus*) | female-Xist Repeat E KO MEF (*M. musculus*/M. castaneus F1 hybrid) | PMID:30827740 | EY.T4-Xist RepE KO | Maintained in Jeannie T Lee lab |
| Cell line (*Mus musculus*) | female-wild type ES (*M. musculus*/M. castaneus F2 hybrid) | PMID:18535243 | TST | Maintained in Jeannie T Lee lab |
| Cell line (*Mus musculus*) | female-Xist Repeat B ES (*M. musculus*/M. castaneus F2 hybrid) | PMID:30827740 | TST-Xist RepB KO | Maintained in Jeannie T Lee lab |
| Cell line (*Mus musculus*) | male-wild type ES (*M. musculus*/M. castaneus F2 hybrid) | PMID:30827740 | J1 | Maintained in Jeannie T Lee lab |
| Commercial assay or kit | NEBNext Ultra II DNA Library Prep Kit for Illumina | NEB | E7645S | NA |
| Commercial assay or kit | NEBNext Ultra II Directional RNA Library Prep Kit for Illumina | NEB | E7760S | NA |
| Commercial assay or kit | NEBNext rRNA Depletion Kit | NEB | E7400L | NA |
| Software, algorithm | trim_galore/cutadapt | https://www.bioinformatics.babraham.ac.uk/projects/trim_galore/ | 0.4.3/1.7.1; RRID:SCR_011847 | NA |
| Software, algorithm | Subread | https://subread.sourceforge.net/ | 2.0.2; RRID:SCR_009803 | RNA-seq counting |
| Software, algorithm | NovoAlign | https://www.novocraft.com/products/novoalign/ | 4.03; RRID:SCR_014818 | CHART/ChIP-seq reads alignment |
| Software, algorithm | Deeptools | https://deeptools.readthedocs.io/en/latest/ | 3.1.2; RRID:SCR_016366 | NA |
| Software, algorithm | MACS2 | https://github.com/macs3-project/MACS | 2.1; RRID:SCR_013291 | NA |
| Software, algorithm | Bedtools | https://bedtools.readthedocs.io/en/latest/ | 2.3; RRID:SCR_006646 | NA |
| Software, algorithm | STAR | https://github.com/alexdobin/STAR | 2.7.10 a; RRID:SCR_004463 | RNA-seq reads alignment |

## Cell lines

The wild-type and *Xist*'s Repeat B or E deletion MEF cell lines (*M. musculus*/M. castaneus F1 hybrid) has been previously described and generated (*Colognori et al., 2019*). Additionally, male *Xist* transgenic (*Xist* TG) MEF cells, previously denoted as '♂X+P' (*Jeon and Lee, 2011*) were included in the study. MEFs were cultured in medium comprising DMEM, high glucose, GlutaMAX Supplement, pyruvate (Thermo Fisher Scientific), 10% FBS (Sigma), 25 mM HEPES pH 7.2–7.5 (Thermo Fisher Scientific), 1 x MEM non-essential amino acids (Thermo Fisher Scientific), 1 x Pen/Strep (Thermo Fisher Scientific), and 0.1 mM βME (Thermo Fisher Scientific), maintained at 37 °C with 5% CO2.

The wild-type and *Xist*'s Repeat B deletion female ES cell line (*M. musculus*/M. castaneus F2 hybrid) harboring a mutated *Tsix* allele has been previously established (*Colognori et al., 2019*; *Ogawa et al., 2008*). The male ES cell line was previously referred to as J1 (*Wang et al., 2018*). ES cells were grown on γ-irradiated MEF feeder cells. Culture conditions included DMEM, high glucose, GlutaMAX Supplement, pyruvate (Thermo Fisher Scientific), 15% Hyclone FBS (Sigma), 25 mM HEPES pH 7.2–7.5, 1 x MEM non-essential amino acids, 1 x Pen/Strep, 0.1 mM βME, and 500 U/mL ESGRO recombinant mouse Leukemia Inhibitory Factor (LIF) protein (Sigma) at 37 °C with 5% CO$_2$.

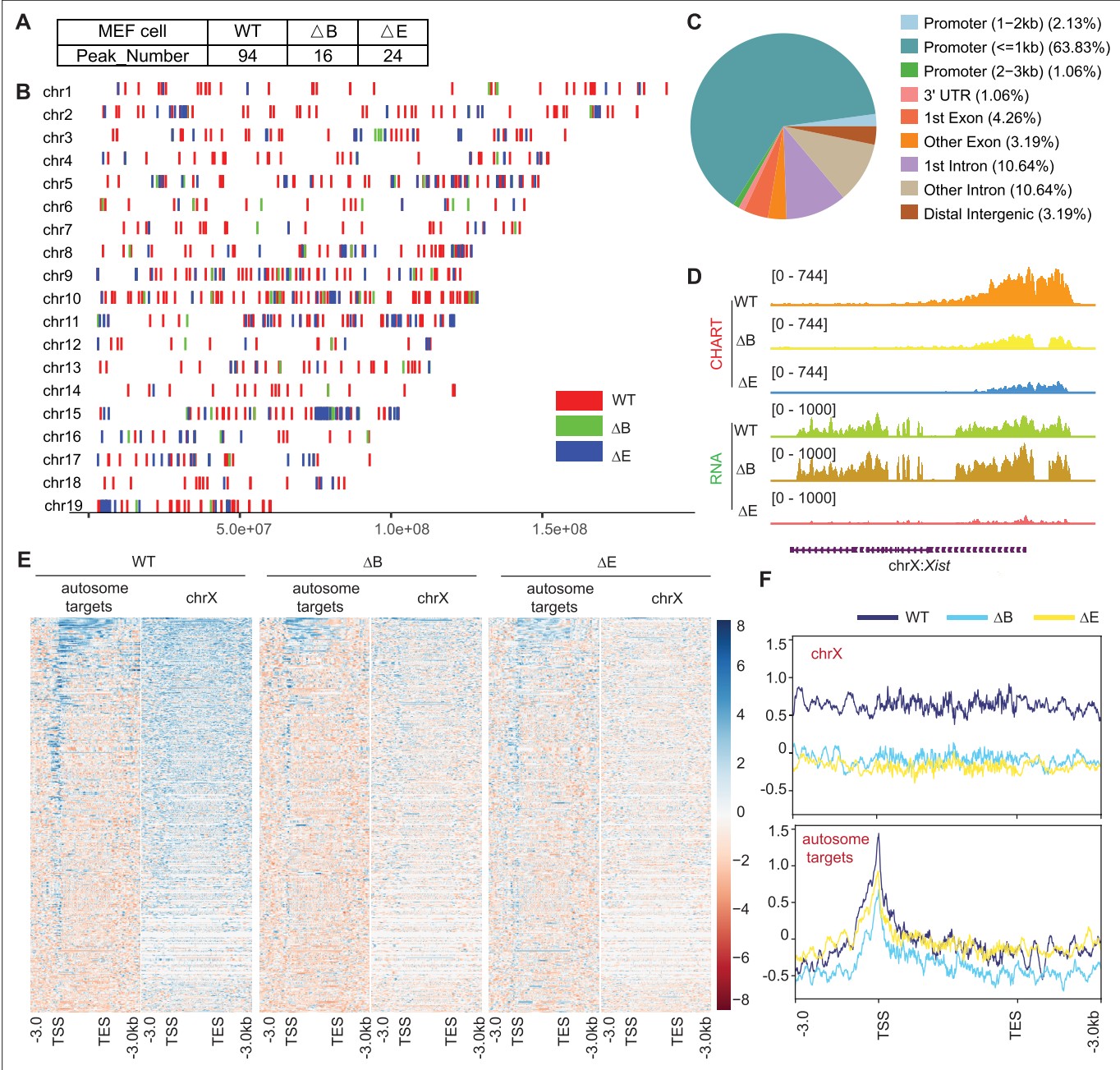

**Figure 6.** Xist RNA also binds autosomal genes in post-X chromosome inactivation (XCI) mouse embryonic fibroblasts (MEF) cells. (**A**) Xist peak number (MACS2 peak calling) on autosomes in wild-type (WT), ΔRepB, and ΔRepE female MEF cells. (**B**) Xist peak patterns (MACS2 peak calling) on autosomes in WT, ΔRepB, and ΔRepE female MEF cells. (**C**) Feature annotation of *Xist* binding loci on autosomes in WT female ES cells. (**D**) Representative capture hybridization analysis of RNA targets (CHART)-seq and RNA-seq track patterns of Xist in WT, ΔRepB, and ΔRepE female MEF cells. (**E–F**) Heatmaps (**E**) and average profiles (**F**) depicting Xist coverage on autosome targets and chromosome X in WT, ΔRepB, and ΔRepE female MEF cells.

The online version of this article includes the following figure supplement(s) for figure 6:

**Figure supplement 1.** *Xist* has overlapped autosomal binding peaks in differentiated embryonic stem (ES) and mouse embryonic fibroblasts (MEF) cells.

## ES cell differentiation

Undifferentiated ES cells were initially cultivated on γ-irradiated MEF feeders for 3 d (day 0). Subsequently, the ES cell colonies were trypsinized (Thermo Fisher Scientific), and the feeders were removed. ES cells were then transitioned to a medium devoid of LIF and cultured in suspension for

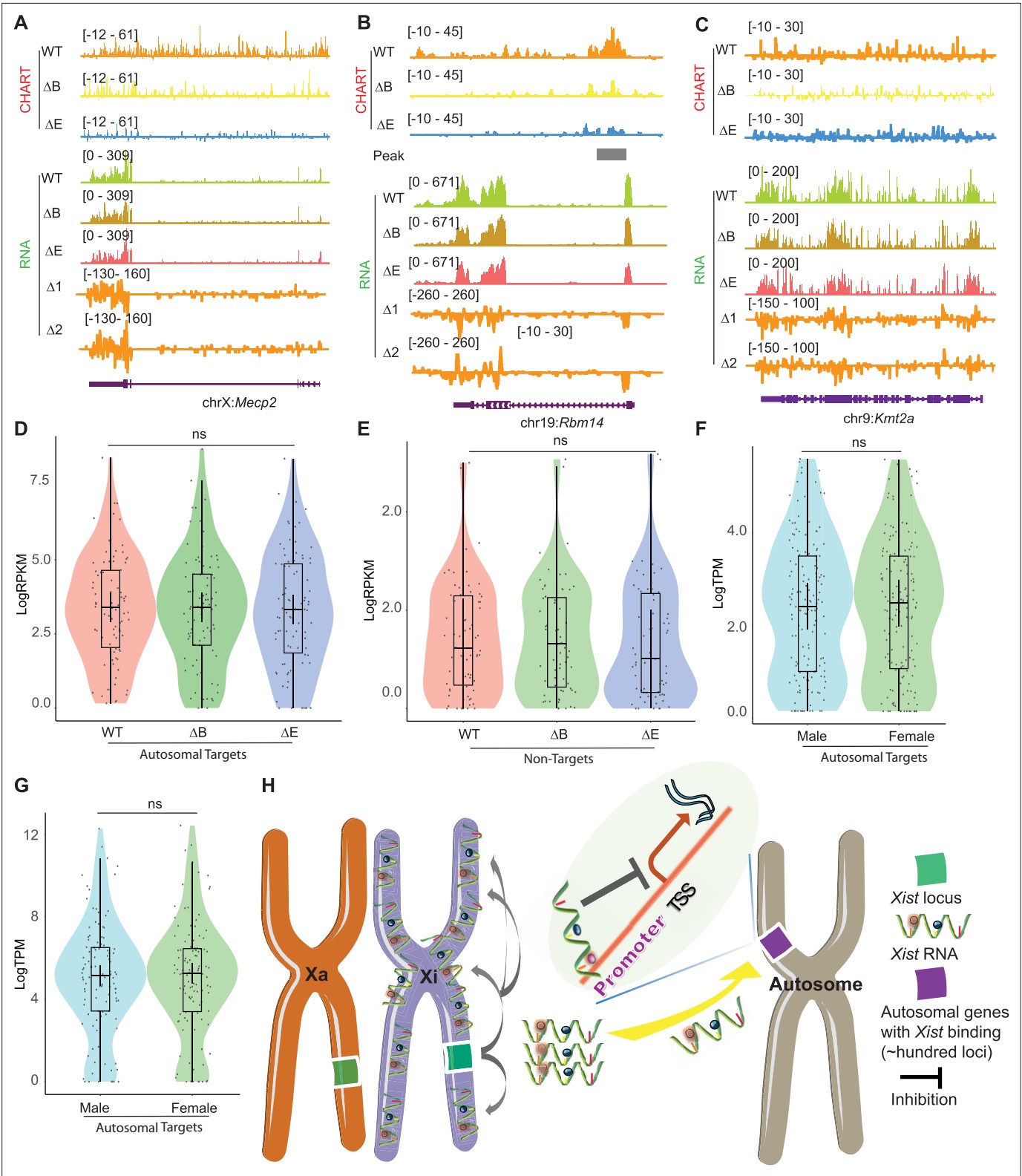

**Figure 7.** *Xist* autosomal binding does not alter gene expression in mouse embryonic fibroblasts (MEF) cells. (**A–C**) Representative capture hybridization analysis of RNA targets (CHART)-seq and RNA-seq track patterns of *Xist* autosomal binding genes such as *Mecp2* (**A**) and *Rbm14* (**B**), and autosome genes without *Xist* binding such as *kmt2a* (**C**) in wild-type (WT), ΔRepB, and ΔRepE female MEF cells. Change in coverage (Δ1 and Δ2) is shown below (Δ1 for ΔRepB -WT, and Δ2 for ΔRepE -WT♀). (**D–E**) Gene expression levels for *Xist* targets (**D**) or non-targets (**E**) on autosomes in WT, ΔRepB, and ΔRepE female MEF cells show no obvious changes. p-values are determined using the Wilcoxon rank sum test. (**F–G**) Gene expression levels for *Xist* targets on

*Figure 7 continued on next page*

*Figure 7 continued*

autosomes in male and female MEF cells show no obvious changes. p-values are determined using the Wilcoxon rank sum test. (**H**) Schematic of the *Xist* autosome binding pattern influences the gene expression. During the differentiation process of female embryonic stem (ES) cells, the *Xist* is specifically expressed from the *Xist* loci of one X chromosome (Xi). It then binds to the *Xist* loci of the Xi in cis, spreading across this chromosome and effectively silencing the expression of most of its genes. Additionally, some *Xist* complexes will also bind to hundreds of loci on autosomes, thereby inhibiting the expression levels of target genes located there.

4 d to form embryoid bodies (EBs). On day 4, the EBs were carefully settled onto gelatin-coated plates and allowed to undergo further differentiation until they were harvested on day 14.

## CHART-seq

*Xist* CHART-seq methodology was conducted in accordance with a previously described protocol (*Wang et al., 2018*). Briefly, the cells were collected and suspended in PBS at a concentration of 2.5 million cells/mL. Subsequently, cross-linking was performed using 1% formaldehyde at room temperature for 10 min, followed by quenching with 0.125 M glycine for an additional 10 min. After two washes with ice-cold PBS, cells were pelleted and snap-frozen in liquid nitrogen. For CHART, 25 million cells were thawed on ice and resuspended in 1 mL ice-cold sucrose buffer (10 mM HEPES pH 7.5, 0.3 M sucrose, 1% Triton X-100, 100 mM potassium acetate, 0.1 mM EGTA, 0.5 mM spermidine, 0.15 mM spermine, 1 mM DTT, 1 x protease inhibitor cocktail, 10 U/mL SUPERase•In RNase Inhibitor). The cell suspension was rotated at 4 °C for 10 min, followed by dilution with 2 mL of cold sucrose buffer. Douncing was performed using an RNaseZAP-treated 15 mL glass Wheaton Dounce tissue grinder 20 times with a tight pestle. The nuclear suspension was carefully layered on top of a cushion of 7.5 ml glycerol buffer (10 mM HEPES pH 7.5, 25% glycerol, 1 mM EDTA, 0.1 mM EGTA, 100 mM potassium acetate, 0.5 mM spermidine, 0.15 mM spermine, 1 mM DTT, 1 x cOmplete EDTA-free protease inhibitor cocktail, and 5 U/mL RNase inhibitor) in a new 15 ml tube, and centrifuged at 1500 g for 10 min at 4 °C. The pellet was resuspended in 3 mL of PBS and cross-linked with 3 ml 6% formaldehyde for 30 min at room temperature with rotation. Afterwards, nuclei were pelleted by centrifugation at 1000 g for 5 min at 4 °C and washed three times with ice-cold PBS, and resuspended in 1 mL ice-cold nuclear extraction buffer (50 mM HEPES, pH 7.5, 250 mM NaCl, 0.1 mM EGTA, 0.5% N-lauroylsarcosine, 0.1% sodium deoxycholate, 5 mM DTT, 10 U/mL RNase inhibitor), rotated for 10 min at 4 °C, and then centrifuged at 400 g for 5 min at 4 °C and resuspended in 230 µL cold sonication buffer (50 mM HEPES pH 7.5, 75 mM NaCl, 0.1 mM EGTA, 0.5% N-lauroylsarcosine, 0.1% sodium deoxycholate, 0.1% SDS, 5 mM DTT, 10 U/mL RNase inhibitor) to a final volume of ~270 µL. The nuclei were sonicated in a microtube using a Covaris E220 sonicator (140 W peak incident power, 10% duty factor, 200 cycles/burst, 4 °C, 300 s). Sonicated chromatin was centrifuged at 16,000 g for 20 min at 4 °C, and transfer the supernatant (~220 µL) to new tube and add 170 µL sonication buffer to a final volume of ~390 µL, which was pre-cleared by 60 uL MyOne Streptavidin C1 beads (Thermo Fisher Scientific) in 640 µL 2 x hybridization buffer (50 mM Tris pH 7.0, 750 mM NaCl, 1% SDS, 1 mM EDTA, 15% formamide, 1 mM DTT, 1 mM PMSF, 1 x cOmplete EDTA-free protease inhibitor cocktail, 100 U/mL RNase inhibitor) at room temperature for 1 hr with rotation. Pre-cleared chromatin was divided into two CHART reactions for *Xist* and control capture, and 1% was saved as an input sample. For each CHART reaction, 36 pmol of antisense (*Xist*) or sense (control) biotinylated capture probes (pooled probe as previously described) were used (*Simon et al., 2013*). Hybridization was performed at room temperature with rotation overnight, and 120 µL C1 beads were added to the samples and incubated at 37 °C for 1 hr with rotation. The beads were washed once with 1 x hybridization buffer (33% sonication buffer, 67% 2 x hybridization buffer) at 37 °C for 10 min, five times with wash buffer (10 mM HEPES pH 7.5, 150 mM NaCl, 2% SDS, 2 mM EDTA, 2 mM EGTA, 1 mM DTT) at 37 °C for 5 min, and twice with elution buffer (10 mM HEPES pH 7.5, 150 mM NaCl, 0.5% NP-40, 3 mM MgCl2, 10 mM DTT) at 37 °C for 5 min. 1% of the final wash was saved as an 'RNA pulldown' sample. CHART-enriched DNA was eluted twice in 200 µL of elution buffer supplemented with 5 U/µL RNase H (New England BioLabs) at room temperature for 20 min. The 'input' sample and CHART DNA were treated with 0.5 mg/mL RNase A (Thermo Fisher Scientific) at 37 °C for 1 hr with rotation and then incubated with 1% SDS, 10 mM EDTA, and 0.5 mg/mL proteinase K (Sigma) at 55 °C for 1 hr. Reverse crosslinking was performed using 150 mM NaCl (final concentration 300 mM) at 65 °C overnight. DNA was purified using phenol-chloroform and further sheared to ~300 bp fragments using Covaris E220e (140 W peak

incident power, 10% duty factor, 200 cycles/burst, 120 s, 4 °C). Sonicated DNA was purified using 1.8 x Agencourt AMPure XP beads (Beckman Coulter). Input and CHART DNA libraries were prepared according to the protocol of the NEBNext Ultra II DNA Library Prep Kit for Illumina (NEB E7645S). Libraries were sequenced on Novaseq S4, generating approximately 30 million 150-nt paired-end reads per sample.

### Data source and processing

Raw data for CHART-seq and bulk RNA-seq generated in this study have been deposited in the Gene Expression Omnibus (GEO) database with accession number GSE271096 and GSE271097, respectively. ChIP-seq data utilized in this study were sourced from a study conducted by David et al. in 2020 (*Colognori et al., 2020*) with GEO accession number: GSE135389. Additionally, RNA-seq and ChIP-seq data specifically pertaining to *Xist* inhibitor X1 were obtained from a dataset published in 2022 (*Aguilar et al., 2022*) with GEO accession number: GSE141683. RNA-seq data for *Xist* TG in mES cells were acquired from another dataset published in 2017 (*Loda et al., 2017*) with GEO accession number: GSE92894. RNA-seq data for male and female MEF cells were acquired from two published datasets with GEO accession numbers: GSE246699 and GSE118443. The data were processed and re-analyzed consistently in this study.

### CHART-seq and ChIP-seq analysis

Initial data preprocessing involved removing adaptors using trim_galore/cutadapt (versions 0.4.3/1.7.1). Subsequently, trimmed reads were prepared for alignment. Alignment of the genomes of *Mus musculus* (mus) and Mus castaneus (cas) was performed using NovoAlign (version 4.03). Aligned reads were then mapped back to the reference mm10 genome using SNPs to generate a BAM file (*Pinter et al., 2012*). Samtools (version 1.11) was employed for random sampling to ensure uniform library sizes across the samples. Utilizing DeepTools (version 3.1.2), input-subtracted ChIP and CHART coverage profiles were created, resulting in bigwig. To discern the peaks of *Xist* CHART on the autosomes, chrX reads were initially excluded from the BAM file. MACS2 (version 2.1) was used for peak calling using default parameters. The resultant peak files from two replicates were merged using bedtools (version 2.30) intersect. Bedtools window (version 2.30) facilitated the creation of gene lists based on peak sizes, spanning 10, 20, 50, and 100 kilobases. Deeptools was used to conduct comparative analyses of coverage, replicate correlation analysis, represented through either profile or heatmap plots, for genomic regions associated with *Xist*, H3K27me3, or H2A119ub across distinct categories of gene lists. IDR (Irreproducible Discovery Rate, version 2.0.2) was used to check the reproducibility of peaks identified in replicates. Intervene (version 6.0.2) was used to generate the Venn diagram of the peak.

### RNA-seq

Total RNA was extracted using TRIzol (Thermo Fisher Scientific), and rRNA was selectively removed using the NEBNext rRNA Depletion Kit (New England BioLabs), according to the manufacturer's instructions. RNA-seq libraries were prepared using the NEBNext Ultra II Directional RNA Library Prep Kit for Illumina (New England BioLabs), which enabled the creation of strand-specific RNA-seq libraries. Libraries were sequenced on Novaseq S4, generating ~30 million 150-nt paired-end reads per sample.

### RNA-seq analysis

Raw RNA-seq data underwent adaptor removal and trimming using trim_galore/cutadapt (versions 0.4.3/1.7.1). RNA-seq data were mapped to three different genomes: C57BL/6 J (mm10), *M. musculus* (mus), and M. castaneus (cas). By employing Subread (version 2.0.2) featureCounts (*Liao et al., 2014*), counts per gene were computed, while deeptools were utilized to generate bigwig files, providing a comprehensive overview of read coverage across the genomes. For expression quantification, both Reads Per Kilobase Million (RPKM) and log-transformed RPKM values were calculated to establish a robust foundation for downstream analysis.

### Quantification and statistical analysis

In the experimental design, two replicates were used for both the CHART and RNA-seq analyses. Statistical analyses were conducted, and the corresponding p-values are transparently reported in the

figures and their respective legends. The significance levels are denoted by asterisks, where *, **, ***, and **** represent $p < 0.05$, $p < 0.01$, $p < 0.001$, and $p < 0.0001$, respectively.

## Acknowledgements

We acknowledge all members of the Lee lab for their invaluable contributions, insightful comments, and stimulating discussions. We thank Dr. Uri Weissbein for his assistance with the CHART protocol. Two NIH grants (R01-HD097665, R01-GM58839) to JTL supported non-overlapping aspects of the study.

## Additional information

### Competing interests

Jeannie T Lee: JTL is an advisor to Skyhawk Therapeutics, a cofounder of Fulcrum Therapeutics, and a non-executive Director of the GSK. The other authors declare that no competing interests exist.

### Funding

| Funder | Grant reference number | Author |
|---|---|---|
| National Institute of General Medical Sciences | R01-HD097665 | Jeannie T Lee |
| National Institute of General Medical Sciences | R01-GM58839 | Jeannie T Lee |

The funders had no role in study design, data collection and interpretation, or the decision to submit the work for publication.

### Author contributions

Shengze Yao, Conceptualization, Data curation, Formal analysis, Validation, Writing - original draft; Yesu Jeon, Conceptualization, Data curation, Formal analysis; Barry Kesner, Formal analysis; Jeannie T Lee, Conceptualization, Formal analysis, Supervision, Funding acquisition, Writing – review and editing

### Author ORCIDs

Shengze Yao (iD) https://orcid.org/0000-0003-4195-8402
Jeannie T Lee (iD) https://orcid.org/0000-0001-7786-8850

Reviewer #1 (Public review): https://doi.org/10.7554/eLife.101197.3.sa1
Reviewer #2 (Public review): https://doi.org/10.7554/eLife.101197.3.sa2
Reviewer #3 (Public review): https://doi.org/10.7554/eLife.101197.3.sa3
Author response https://doi.org/10.7554/eLife.101197.3.sa4

## Additional files

### Supplementary files

Supplementary file 1. *Xist* binding peaks on autosomal regions (MACS2 peak calling) information at day 4 in WT female mouse ES cells.

Supplementary file 2. *Xist* binding peaks on autosomal regions (MACS2 peak calling) information at day 7 in WT female mouse ES cells.

Supplementary file 3. *Xist* binding peaks on autosomal regions (MACS2 peak calling) information at day 14 in wild-type (WT) female mouse embryonic stem (ES) cells.

Supplementary file 4. *Xist* target genes on autosomal regions (10 kb among the binding peak) information at day 4 in wild-type (WT) female mouse embryonic stem (ES) cells.

Supplementary file 5. *Xist* target genes on autosomal regions (10 kb among the binding peak) information at day 7 in wild-type (WT) female mouse embryonic stem (ES) cells.

Supplementary file 6. *Xist* target genes on autosomal regions (10 kb among the binding peak) information at day 14 in WT female mouse ES cells.

Supplementary file 7. *Xist* binding peaks on autosomal regions (MACS2 peak calling) information on wild-type (WT) female mouse embryonic fibroblasts (MEFs).

Supplementary file 8. *Xist* binding peaks on autosomal regions (MACS2 peak calling) information on ΔRepB female mouse embryonic fibroblasts (MEFs).

Supplementary file 9. *Xist* binding peaks on autosomal regions (MACS2 peak calling) information on ΔRepE female mouse embryonic fibroblasts (MEFs).

Supplementary file 10. *Xist* target genes on autosomal regions (10 kb among the binding peak) information on wild-type (WT) female mouse embryonic fibroblasts (MEFs).

Supplementary file 11. Statistical information in this manuscript.

MDAR checklist

## Data availability

Raw data for CHART-seq and bulk RNA-seq generated in this study have been deposited in the Gene Expression Omnibus (GEO) database with accession number GSE271096 and GSE271097, respectively. ChIP-seq data utilized in this study were sourced from a study conducted by David et al. in 2020 33 with GEO accession number: GSE135389. Additionally, RNA-seq and ChIP-seq data specifically pertaining to Xist inhibitor X1 were obtained from a dataset published in 2022 36 with GEO accession number: GSE141683. RNA-seq data for Xist TG in mES cells were acquired from another dataset published in 2017 35 with GEO accession number: GSE92894. RNA-seq data for male and female MEF cells were acquired from two published dataset with GEO accession number: GSE246699 and GSE118443. The data were processed and re-analyzed consistently in this study.

The following datasets were generated:

| Author(s) | Year | Dataset title | Dataset URL | Database and Identifier |
|---|---|---|---|---|
| Yao S, Jeon Y, Kesner B, Lee JT | 2024 | Xist RNA targets selective autosomal genes to modulate expression in a Repeat B-dependent manner [CHART-seq] | https://www.ncbi.nlm.nih.gov/geo/query/acc.cgi?acc=GSE271096 | NCBI Gene Expression Omnibus, GSE271096 |
| Yao S, Jeon Y, Kesner B, Lee JT | 2024 | Xist RNA targets selective autosomal genes to modulate expression in a Repeat B-dependent manner [RNA-seq] | https://www.ncbi.nlm.nih.gov/geo/query/acc.cgi?acc=GSE271097 | NCBI Gene Expression Omnibus, GSE271097 |

The following previously published datasets were used:

| Author(s) | Year | Dataset title | Dataset URL | Database and Identifier |
|---|---|---|---|---|
| Kim J, Mizukami H, Fukamizu A | 2018 | Identification of differentially expressed genes between male and female in mouse embryonic fibroblasts (MEFs). | https://www.ncbi.nlm.nih.gov/geo/query/acc.cgi?acc=GSE118443 | NCBI Gene Expression Omnibus, GSE118443 |
| Beluzic R, Skrinjar I, Pinteric M, Hadzija M, Simunic E, Balog T, Sobocanec S | 2024 | Transcriptomic analysis of Sirt3 knock-out mouse embryonic fibroblasts reveals major differences in metabolism and stress-response between sexes | https://www.ncbi.nlm.nih.gov/geo/query/acc.cgi?acc=GSE246699 | NCBI Gene Expression Omnibus, GSE246699 |
| Loda A, Vassilev I, Brandsma J, van IJcken W, Heard E, Gribnau J | 2017 | The efficiency of Xist-mediated silencing of X-linked and autosomal genes is determined by the genomic environment | https://www.ncbi.nlm.nih.gov/geo/query/acc.cgi?acc=GSE92894 | NCBI Gene Expression Omnibus, GSE92894 |

*Continued on next page*

*Continued*

| Author(s) | Year | Dataset title | Dataset URL | Database and Identifier |
|---|---|---|---|---|
| Aguilar R, Spencer KB, Kesner B, Rizvi NF, Badmalia MD, Mrozowich T, Mortison JD, Rivera C, Smith GF, Burchard J, Dandliker P, Patel TR, Nickbarg EB, Lee JT | 2020 | Targeting Xist with compounds that disrupt RNA structure and X-inactivation | https://www.ncbi.nlm.nih.gov/geo/query/acc.cgi?acc=GSE141683 | NCBI Gene Expression Omnibus, GSE141683 |
| Colognori D, Sunwoo H, Wang D, Wang CY, Lee JT | 2020 | Xist Repeats A and B account for two distinct phases of X-inactivation establishment | https://www.ncbi.nlm.nih.gov/geo/query/acc.cgi?acc=GSE135389 | NCBI Gene Expression Omnibus, GSE135389 |

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
