## [Editor Report · eLife Assessment]

This **valuable** study addresses the potential roles of the master regulator of X chromosome inactivation, the Xist long non-coding RNA, in the regulation of autosomal genes. Using data from mouse cells, the authors propose that Xist can coat specific autosomal promoters, which in turn leads to the attenuation of their transcriptional activity. The evidence from individual genes is interesting, and the model aligns with recently published results from humans. However, despite some improvements during revision, the data and statistical analyses in the current study are not yet strong enough to allow for conclusive inferences, leaving the evidence for mouse cells behaving like human cells **incomplete**. The topic of the work is of broad interest, in particular to colleagues studying gene regulation and noncoding RNAs.

---

## [Referee Report · Reviewer #1 (Public review)]

Summary:

The manuscript by Yao S. and colleagues aims to monitor the potential autosomal regulatory role of the master regulator of X chromosome inactivation, the Xist long non-coding RNA. It has recently become apparent that in the human system, Xist RNA can not only spread in cis on the future inactive X chromosome but also reach some autosomal regions where it recruits transcriptional repression and Polycomb marking. Previous work has also reported that Xist RNA can show a diffused signal in some biological contexts in FISH experiments.

In this study, the authors investigate whether Xist represses autosomal loci in differentiating female mouse embryonic stem cells (ESCs) and somatic mouse embryonic fibroblasts (MEFs). They perform a time course of ESC differentiation followed by Capture Hybridization of Associated RNA Targets (CHART) on both female and male ESCs, as well as pulldowns with sense oligos for Xist. The authors also examine transcriptional activity through RNA-seq and integrate this data with prior ChIP-seq experiments. Additional experiments were conducted in MEFs and Xist-ΔB repeat mutants, the latter fails to recruit Polycomb repressors.

Based on this experimental design, the authors make several bold claims:

(1) Xist binds to about a hundred specific autosomal regions.

(2) This binding is specific to promoter regions rather than broad spreading.

(3) Xist autosomal signal is inversely correlated with PRC1/2 marks but positively correlated with transcription.

(4) Xist targeting results in the attenuation of transcription at autosomal regions.

(5) The B-repeat region is important for autosomal Xist binding and gene repression.

(6) Xist binding to autosomal regions also occurs in somatic cells but does not lead to gene repression.

Together, these claims suggest that Xist might play a role in modulating the expression of autosomal genes in specific developmental and cellular contexts in mice.

Strengths:

This paper deals with an interesting hypothesis that Xist ncRNA can also function at autosomal loci.

Weaknesses:

The revised manuscript now includes many additional bioinformatic analyses to support the premise that Xist RNA targets a specific set of about 100 promoters and attenuates their expression in the early stages of differentiation. I have previously raised significant concerns about the bioinformatic analyses and the robustness of the data, especially those linked to CHART-seq datasets. Despite some improvements, fundamental problems with the analysis remain, precluding a conclusion on whether Xist RNA binds specific autosomal promoters. The main concerns include:

(1) The authors nicely explain the use of biological replicates; however, they still fail to provide the sufficient analysis I requested on d0 and sense probes. While some quantification is presented in Figures 1E and 1F, the peak calling I asked for has still not been performed. In the response document, the authors report that about 600 peaks were identified in d0 female ESCs compared to about 100 in differentiated conditions. They explain this by the well-known phenomenon of having a background of differentiated cells in d0. In my opinion, this reasoning is flawed. With 98% of cells not inducing Xist in the culture, it is unimaginable why 600 peaks would be detected in the peak calling analysis. Rather, this demonstrates a high background in the CHART peak calling. To assess this further, I have reanalyzed d7 CHART datasets and found robust enrichment of the sense probe on promoters of genes, even stronger than the antisense probe. MACS peak calling also identifies a robust number of peaks on the sense probe. Indeed, even though Figure 1F shows low sense probe enrichment, this is because it focuses on the anti-sense peaks only. An opposite effect is observed when focusing on all genes or on sense peaks. Thefore it is tough to decide which of the signal is truelly due to Xist binding and what is an inherent problem with the CHART signal. These results cast serious doubts on the biological conclusions of this work and point to a very high background level of promoter signal in both sense and antisense samples.

(2) The authors do not address the conundrum of their results: how is it possible to have a genome-wide autosomal accumulation of Xist signal at promoters (see Figures 1A and 1B), while simultaneously specifically affecting only 100 promoters in the genome? The signal is either general (as Figures 1A and 1B suggest) or specific (as implied by the peak calling), but it cannot be both. Current data points to the fact that CHART has a bias for the most open parts of the chromatin.

(3) The text is still very confusing when it comes to Polycomb. Some experiments point to the fact that there are few PRC1/2 marks at putative Xist autosomal binding sites (Figure 3C), while the use of X1 induces the loss of PRC2 marks. I still find this internally contradictory. The authors sadly do not address my concerns with additional analysis. Their current data indicate that upon Xist upregulation, Xist-RNA binds to autosomal regions that are highly expressed and devoid of Polycomb. These loci then become transcriptionally attenuated and gain some (but low) level of PRC2 in a Xist-dependent fashion. If this model is true, then all these regions should not have Xist in d0 of differentiation and should also have slightly lower levels of PRC2. The argument that there is a low level of Xist in 2-5% of cells should not be a problem because most of the signal will come from the 98% of cells not expressing Xist (as seen in Figure 1A). Without timepoint 0, the whole premise of the paper is difficult to interpret. Either the d0 samples are good enough, or the system is so leaky that it is nearly impossible to identify Xist-specific effects. Males are a useful control but are obviously a genetically very different line with distinct epigenetic and signaling statuses. It is crucial to compare the timing of repression/PRC accumulation to conclude if and how Xist is functional on these loci.

(4) The authors did not address my concerns about the transcriptional analysis. I belive that the control genes are not selected properly. This analysis should not have been performed on just 100 randomly selected regions/genes. Instead, bootstrapping of 100 randomly selected regions/genes should be done, e.g., 1000 times. Additionally, one should only sample from expressed genes to have a comparable control gene set. For example, in Figures 4D and 4E, the distribution of control regions is entirely different. To stress again, relying on a set of 100 randomly selected genes/regions is not statistically robust; controls have to be matched, and bootstrapping has to be performed. Finally, each timepoint uses a different set of autosomal targets. There is a need to visualize the same set of genes across all timepoints (including d0). For example, are genes bound by Xist at d7 highly expressed at d0 and then attenuated only at d7? What happens to them at d14 (see points from 3)? The arguments about d0 heterogeneity are again not convincing (nor is Figure 3H, which shows a different set of genes).

(5) Transcriptional analysis is often shown only as tracks however the reads for key example genes have to be quantified properly and not just visualized or amalgamated in a violin plot.

---

## [Referee Report · Reviewer #2 (Public review)]

Summary:

To follow-up on recent reports of Xist-autosome interaction the authors examine female (and male transgenic) mESCs and MEFs by CHARTseq. Upon finding that only 10% of reads map to X, they sought to identify reproducible alternative sites of Xist-binding, and identify ~100 autosomal Xist-binding sites in active chromatin regions. They demonstrate a transient down-regulation of autosomal expression. They utilize published male transgenic inducible Xist mESC data to support their findings. In their system, inhibition of Xist reduces autosomal impact.

Strengths:

The authors address a topical and interesting question with a series of models including developmental timepoints and utilize unbiased approaches (CHARTseq, RNAseq). For the CHARTseq they have controls of both sense probes and male cells; and indeed do detect considerable background with their controls. The use of 'metagene' plots provides a visual summation of genic impact. They compare with published data.

Weaknesses:

The revised text and rebuttal clarified my confusion of the 'follow-up' analyses (Figure 4) compared to published datasets. Further, the figure legends have been improved.

While the controls were a strength, it appears that when focussed on bound regions, the background (from sense probes) is now also substantially higher than global background (compare 1E to 1A/B). Thus, why do these autosomal targets enrich for the sense probes, and how to distinguish from such background for the ∆B experiments? If male and sense are both controls, then why is sense lower for males than females, doesn't this suggest Xist impact? While authors note d0 might detect Tsix, the signal is only slightly reduced by d14 and never equivalent. Indeed, the new PCA (S1C) does show as noted that female Xist interactions are distinct from sense and male, but the male signal is even more distinct from sense probes.

It would have been preferable to see the dispersion of the Xist RNA cloud in these ∆B cells, rather than a reference.

Only 2 replicates were used, but there were multiple time-points: D0, D4, d7, d14; further, the correlation analysis showed good reproducibility, and in response to reviews they note that 2 replicates are standard of practice.

The conclusion that RepB is "required for localization to the ~100 genes" is based on density (panel 2E); however, these autosomal targets retain enrichment at TSSs (panel 2A) and indeed the text suggests they are the same sites, suggesting that in fact the choice of autosomal region binding is not RepB dependent. Thus, this remains unresolved for me.

The introduction is clear, and the senior author is a leader in the field; however, by this reviewer's count 19 of the 52 references include the senior author.

Better descriptors for the supplemental Excel files would be helpful.

Aim achievement: The authors do identify autosomal sites with enrichment of chromatin marks and evidence of silencing. Their revised text clarifies many issues, although this reviewer still remains unconvinced that the autosomal targeting is repB-dependent.

The impact of Xist on autosomes is important for consideration of impact of changes in Xist expression with disease (notably cancers). Knowing the targets (if consistent) would enable assessment of such impact.

---

## [Referee Report · Reviewer #3 (Public review)]

Summary:

Yao et al use CHART to identify chromatin associated with Xist in female mouse ESCs, and, as control, male ESCs at various timepoints of differentiation. Besides binding of Xist to X chromosome regions they found significant binding to autosomes, concentrating mostly on promoter regions of around 100 autosomal genes, as elucidated by MACS. The authors went on to show that the RepB repeat is mostly responsible for these autosomal interactions using a female ESC line in which RepB is deleted. Evidence is provided that Xist interacts with active autosomal genes containing lower coverage of repressive marks H3K27me3 and H2AK119ub and that RepB dependent Xist binding leads to dampening of expression, but not silencing of autosomal genes. These results were confirmed by overexpression studies using transgenic ESCs with doxycycline-inducible Xist as well as via a small molecule inhibitor of Xist (X1), inducing/inhibiting the dampening of autosomal genes, respectively. Finally, using MEFs and Xist mutants RepB or RepE the authors provide evidence that Xist is bound to autosomal genes in cells after the XCI process but appears not to affect gene expression. The data presented appear generally clear and consistent and indicate some differences between human and mouse autosomal regulation by Xist. Thus, these results are timely and should be published.

Strengths:

Regulation of autosomal gene expression by Xist is a "big deal" as misregulation of this lncRNA causes developmental defects and human disease. Moreover, this finding may explain sex-specific developmental differences between the sexes. The results in this manuscript identify specific mouse autosomal genes bound by Xist and decipher critical Xist regions that mediate this binding and gene dampening. The methods used in this study are appropriate, and the overall data presented appear convincing and are consistent, indicating some differences between human and mouse autosomal regulation by Xist.

Comments on revisions:

In the revised manuscript, the authors have addressed my previous criticisms satisfactorily. Moreover, the manuscript has been much improved with new confirmatory results and additional control experiments. This, combined with more detailed descriptions/explanations facilitates data interpretation, making the paper more transparent and easier to read.

---

## [Author Response]

The following is the authors’ response to the original reviews.

**Reviewer #1 (Public review):**
Summary:The manuscript by Yao S. and colleagues aims to monitor the potential autosomal regulatory role of the master regulator of X chromosome inactivation, the Xist long non-coding RNA. It has recently become apparent that in the human system, Xist RNA can not only spread in cis on the future inactive X chromosome but also reach some autosomal regions where it recruits transcriptional repression and Polycomb marking. Previous work has also reported that Xist RNA can show a diffused signal in some biological contexts in FISH experiments.In this study, the authors investigate whether Xist represses autosomal loci in differentiating female mouse embryonic stem cells (ESCs) and somatic mouse embryonic fibroblasts (MEFs). They perform a time course of ESC differentiation followed by Capture Hybridization of Associated RNA Targets (CHART) on both female and male ESCs, as well as pulldowns with sense oligos for Xist. The authors also examine transcriptional activity through RNA-seq and integrate this data with prior ChIP-seq experiments. Additional experiments were conducted in MEFs and Xist-ΔB repeat mutants, the latter fails to recruit Polycomb repressors.Based on this experimental design, the authors make several bold claims:(1) Xist binds to about a hundred specific autosomal regions.(2) This binding is specific to promoter regions rather than broad spreading.(3) Xist autosomal signal is inversely correlated with PRC1/2 marks but positively correlated with transcription.(4) Xist targeting results in the attenuation of transcription at autosomal regions.(5) The B-repeat region is important for autosomal Xist binding and gene repression.(6) Xist binding to autosomal regions also occurs in somatic cells but does not lead to gene repression.Together, these claims suggest that Xist might play a role in modulating the expression of autosomal genes in specific developmental and cellular contexts in mice.Strengths:This paper deals with an interesting hypothesis that Xist ncRNA can also function at autosomal loci.Weaknesses: The claims reported in this paper are largely unsubstantiated by the data, with multiple misinterpretations, lacking controls, and inadequate statistics. Fundamental flaws in the experimental design/analysis preclude the validity of the findings. Major concerns are listed below: (1) The entire paper is based on the CHART observation that Xist is specifically targeted to autosomal promoters. Overall, the data analysis is flawed and does not support such conclusions. Importantly the sense WT and the 0h controls are not used, nor are the biological replicates.

We respectfully disagree with Rev1 but nevertheless thank the reviewer for making some suggestions that helped to strengthen our manuscript. We have provided new experiments and analyses in the revised manuscript. Please see responses below.

Rev1 seems to have missed or misunderstood some key experiments. In fact, the sense WT and 0 hr controls were shown. Furthermore, we included at least two biological replicates for each experiment.

We used both male ES cells (which do not express Xist) and sense probes as key negative controls, as outlined in Figure S1. Crucially, we only analyzed peaks that were reproducible between biological replicates. The Xist CHART peaks in differentiating female ES cells were significantly enriched above the “background” defined by the sense probe and male controls. Specifically, in comparison to undifferentiated female ES cells (day 0) where both X chromosomes are active and Xist is not induced, Xist CHART robustly pulled down the X chromosome during cell differentiation (day 4, day 7, and day 14). In contrast, male ES cells showed no significant pull-down of the X chromosome, and the sense group also exhibited markedly reduced binding (new Figure S1B). Furthermore, Principal Component Analysis (PCA) of CHART-seq reads (day 4 as an example) include Xist, sense, and input in WT and ΔRepB female, further confirmed that the sense probe CHART was clearly distinguishable from Xist CHART signals. Please see revised Figure S1C. Together, these findings underscore the specificity and robustness of our CHART results.

Data is typically visualized without quantification, and when quantified, control loci/gene sets are erroneously selected. Firstly, CHART validation on the X in FigS1 is misleading and not based on any quantifications (e.g., see the scale on Kdm6a (0-190) compared to Cdkl5 (0-40)). If scaled appropriately, there is Xist signal on the escapee.

Rev1 may have misread the presented data. In the example raised by Rev1, Fig. S1 is inherently quantitative: e.g., a ratio is a number in Fig. S1A (now Fig. S1B) and all gene tracks in Fig. 1B-E are shown with scales. We showed X-linked genes in Fig. S1 (now Fig. S2) as a control to demonstrate that the CHART worked and that Xist accumulated over time from day 0 to day 14. Our new Figure 1B demonstrates the Xist accumulation in graph format.

Our paper focuses on Xist autosomal binding sites. Thus, the X-linked examples were placed in the supplement. Escapee genes do in fact accumulate Xist at their promoter regions and this finding is consistent with data published by Simon et al. (2013, Nature). It was therefore not desirable in this paper to reanalyze X-linked genes, including escapees. Nevertheless, to address the reviewer’s concerns, we present new data in new Figure S3A. Here we analyzed the density of Xist binding across X-linked genes, including both active and inactive genes, as well as escapee genes. From this quantitative analysis, it should be clear that escapees do bind Xist. However, from the metagene plots in Figure S3B, we confirm the previous conclusion that escapees bind Xist at high levels just upstream of the promoter and that there is a depletion of Xist in the escapee gene body, consistent with a barrier preventing Xist from moving into the active gene.

All X-linked loci should have been quantified and classified based on escape status; sense control should also be quantified, and biological replicates should be shown separately.

Please see above response.

Additionally, in the revised manuscript, we have examined the Irreproducible Discovery Rate (IDR) to validate the reproducibility of peaks between the two replicates in the revised version, and we included a representative example from female WT ES cells at day 4 (revised Figure S4A). The results showed a strong correlation between the replicates, with an IDR threshold of 0.05 (red point > 0.05). As described in the Methods section, to ensure reliable and robust peak identification, we performed peak calling (MACS2) separately on each replicate, and then used *bedtools intersect* to identify peaks that overlapped between the two replicates. This stringent process, including strict q-value settings in MACS2, ensures the reliability and reproducibility of the peaks presented in this study.

Secondly, and most importantly, Figure 1 does not convincingly show specific Xist autosomal binding. Panel A quantification is on extremely variable y-scales and actually shows that Xist is recruited globally to nearly all autosomal genes, likely indicating an unspecific signal. Again, the sense and 0h controls should have been quantified along with biological replicates.

Figure 1 shows heatmaps and corresponding metagenes for d0, d4, d7, and d14 female ES cells. Two biological replicates are analyzed. In our revised manuscript, we have used Pearson and Spearman correlation coefficients to measure the strength and direction of a relationship between two biological replicates and shown that the two replicates have high reproducibility (new Figure S1A). On d0, the Xist coverage on autosomes and X chromosome is low, but there is a clear increase on d4, d7, and d14, particularly at the TSS of autosomal genes, as shown by the metagene plots on in Figure 1A-B and the CHART density maps in new Figure 1E-F. We also show relative depletion of Xist signals in the male and sense negative controls.

Upon inspecting genome browser tracks of all regions reported in the manuscript (Rbm14, Srp9, Brf1, Cand2, Thra, Kmt2c, Kmt2e, Stau2, and Bcl7b), the signal is unspecific on all sites with the possible exception of Kmt2e. On all other loci, there is either a strong signal in the 0h ESC controls or more signal in some of the sense controls. This implies that peak calling is picking up false positive regions. How many peaks would have been picked up if the sense or the 0h controls were used for peak calling? It is likely that there would be a lot since there are also possible "peaks" (e.g., Fzd9) in control tracks.

The analysis cannot be performed by visual inspection. A statistical analysis must be performed to call signal above noise. This is why we performed peak-calling on two biological replicates and identified overlapping peaks using *bedtools intersect* to improve reliability. Significant peaks are noted as black bars under each track. As mentioned above, for our analysis, we focused on the top 100 peaks based on peak scores to ensure robustness. Xist has significantly higher signal compared to the sense probe in the Xist-autosomal peak regions (revised Figure 1E-F). Additionally, we conducted peak calling on undifferentiated ES cells (d0) and detected a significantly higher number of peaks (~600) compared to the differentiated states (d4 or d7) (~100).

Single-cell sequencing studies have shown that about 2% of undifferentiated mESCs express detectable Xist (Pacini et al., *Nat Commun*, 2021). The Xist peaks in “day 0” cells may be due to the differentiating population.

Further inspection of the data was not possible as the authors did not provide access to the raw fastq files. When inspecting results from past published experiments {Engreitz, 2013 #1839} reported regions were not bound by Xist.

On the contrary, we deposited the raw data files to GEO prior to the submission of the paper and included the reviewer link to access them. As of August 24, 2024, GEO publicly released these files, allowing for full inspection of the data.

Regarding the Engreitz publication, it is not recommended to compare our current study to their analysis for the crucial reason that the Engreitz study was not conducted under physiological conditions. The authors overexpressed the *Xist* gene in male ES cells. Because Xist RNA can silence genes in male cells as well, this ectopic overexpression normally leads to cell death — thus forcing examination of effects in a narrow time window before Xist can fully spread and act across the genome. Comparing our experiments (endogenous Xist expression in female ES cells) to the ectopic overexpression in male ES cells of Engreitz et al. should therefore not be undertaken.

Thirdly, contrary to the authors' claim, deleting the B repeat does not lead to a loss of autosomal signal. Indeed, comparing Fig1A and Fig2B side by side clearly shows no difference in the autosomal signal, likely because the autosomal signal is CHART background. Properly quantifying the signal with separate replicates as well as the sense and 0h controls is vital. Overall current data together with published results indicate that CHART peak calling on autosomes is due to technical noise or artefacts.

In our revised manuscript, we have included the quantitative results as mentioned above in the main and supplementary figure (new Figure 1E-F, Figure 2E-F, and S3A). The data clearly show an enrichment in the Xist CHART samples in differentiating female ES cells.

We believe the reviewer may be comparing the original Figure 1A and Figure 2A (not Figure 2B). As mentioned above, the analysis cannot be performed by visual inspection. Please see new Figure 2E and 2F. From these data, it should be clear that deleting RepB causes a decrease in Xist targeting to autosomal loci.

(2) The RNA-seq analysis is also flawed and precludes strong statements. Firstly, the analysis frequently lacks statistical analysis (Fig3B, FigS2B-C) and is often based on visualizations (Fig 3D-G) without quantifications. Day 4 B-repeat deletion does not lead to a significant change in the expression of genes close to Xist signal (Fig3H, d14 does not fully show).

Please see new revised Figure 3B and Figures S2B-C (now revised as Figures S6A and S6B).

Secondly, for all transcriptional analysis, it is important to show autosomal non-target genes, which is not always done.

In the revised manuscript, we included non-target genes for each analysis (new Figure 4E-F, 5D and 5F, 7C and 7E, S7F, S8).

Indeed, both males and B repeat deletion will lead to transcriptional changes on autosomes as a secondary effect from different X inactivation status. The control set, if used, is inappropriate as it compares one randomly selected set of ~100 genes. This introduces sampling error and compares different classes of genes. Since Xist signal targets more active genes, it is important to always compare autosomal target genes to all other autosomal genes with similar basal expression patterns.

Please see new Figure S8. We included 100 randomly selected non-target sites on autosomes for this comparative analysis. For consistency, we applied the same flanking regions (10 kb) in the analysis of both target and non-target genes. We believe that this selection method for nontargets is appropriate for two reasons: first, it allows us to control for Xist binding and non-binding; second, it ensures a similar number of genes in both groups, providing a robust foundation for statistical analysis.

(3) The ChIP-seq analysis also has some problems. The authors claim that there is no positive correlation between genes close to Xist autosomal binding (10kb) compared to those 50kb away (Fig 3C, S2D); however, this analysis is based entirely on metagene visualization. Signal within the Xist binding sites should be quantified (not genes close by) and compared to other types of genomic loci and promoters. Focusing on the 50kb group only as controls is misleading.

We believe the reviewer may have misunderstood our conclusions. As stated in the paper, we observed lower coverage of the histone marks H3K27me3 and H2AK119ub, associated with PRC2 and PRC1, respectively. Our conclusions regarding PRC1/2 support the RNA-seq results, indicating that Xist tends to bind to actively expressed genes. In other words, these genes exhibit lower levels of PRC-mediated silencing signals. This observation underscores the relationship between Xist binding and gene activity, highlighting that Xist preferentially associates with regions that are less subject to silencing by polycomb repressive complexes.

Secondly, the authors only look at PRC mark signal upon differentiation; what about the 0h timepoint, i.e., is there pre-marking?

Day 0 is not an appropriate timepoint for this analysis because Xist is not yet induced. There is also a small fraction of cells (<5%) that spontaneously differentiate and start to undergo XCI. Because of these reasons, the day 0 timepoint is considered somewhat heterogeneous and it would be difficult to make conclusions regarding Xist peaks in these samples.

Most worryingly, the data analysis is not consistent between figures (see Fig3C vs 5H-I). In Fig5, the group of Xist targets was chosen as those within 100kb of Xist binding, which would encompass all the control regions from Fig3C. In this analysis, the authors report that there is Xist-dependent H3K27me3 deposition, and in fact, here the Xist autosomal targets have more of it than the controls. Overall, all of this analysis is misleading, and clear conclusions cannot be made.

We believe that the reviewer may have also misunderstood the analysis in Figure 5. Figure 5 shows the effect of the Xist inhibitor, X1, on H3K27me3 and gene expression. X1 blocks reduces PRC2 targeting and gene silencing — consistent with X1’s effect on RepA as published in Aguilar et al. 2022.

All in all, because the fundamental observation is not robust (see point 1), all subsequent analyses are also affected. There are also multiple other inconsistencies within the analysis; however, they have not been included here for brevity.

We again respectfully disagree with Rev1 but thank the reviewer for making suggestions that helped to strengthen our manuscript. We believe that the revised manuscript with new analyses is improved in part because of the reviewer’s critical comments.

**Reviewer #2 (Public review):**
Summary:To follow-up on recent reports of Xist-autosome interaction the authors examine female (and male transgenic) mESCs and MEFs by CHARTseq. Upon finding that only 10% of reads map to X, they sought to identify reproducible alternative sites of Xist-binding, and identify ~100 autosomal Xistbinding sites and show a transient impact on expression.Strengths:The authors address a topical and interesting question with a series of models including developmental timepoints and utilize unbiased approaches (CHARTseq, RNAseq). For the CHARTseq they have controls of both sense probes and male cells; and indeed do detect considerable background with their controls. The use of deletions emphasizes that intact functional Xist is involved. The use of 'metagene' plots provides a visual summation of genic impact.

Reviewer 2 has made some excellent suggestions. We have revised the manuscript accordingly and are grateful to the reviewer for the recommendations.

Weaknesses:Overall, the result presentation has many 'sample' gene presentations (in contrast to the stronger 'metagene' summation of all genes). The manuscript often relies on discussion of prior X chromosomal studies, while the data generated would allow assessment of the X within this study to confirm concordance with prior results using the current methodology/cell lines.Many of the 'follow-up' analyses are in fact reprocessing and comparison of published datasets. The figure legends are limited, and sample size and/or source of control is not always clear. While similar numbers of autosomal Xist-binding sites were often observed, the presented data did not clarify how many were consistent across time-points/cell types. While there were multiple time points/lines assessed, only 2 replicates were generally done.

We apologize for the deficiencies in the legend. The revised manuscript has corrected them.

We generated many new datasets with deep sequencing, with at least two biological replicates for each. Such experiments are extremely expensive by nature. Thus, two biological replicates are typically considered acceptable.

Additionally, we performed reanalysis of published datasets to test whether — *in the hands of other investigators* — cell lines expressing Xist also supported autosomal targeting. Figure 4 is a case in point. Here we examined Tg1 and Tg2, which respond to doxycycline to overexpress Xist from an ectopic site. Transcriptomic analysis showed significant downregulation of autosomal Xist targets, as exemplified by *Rbm14* and *Bcl7b* (new Figure 4C, S9B). In contrast, non-targets of Xist such as *Stau1* did not demonstrate significant changes in gene expression (new Figure 4E and 4G). Looking across all autosomal target genes, we observed a significant decrease in mean expression in the Xist overexpressing cell lines (new Figure 4D). The fact that the autosomal changes were also observed in datasets generated by other investigators greatly strengthen our conclusions.

Aim achievement:The authors do identify autosomal sites with enrichment of chromatin marks and evidence of silencing. More details regarding sample size and controls (both treatment, and most importantly choice of 'non-targets' - discussed in comments to authors) are required to determine if the results support the conclusions.Specific scenarios for which I am concerned about the strength of evidence underlying the conclusion:I found the conclusion "Thus, RepB is required not only for Xist to localize to the X- chromosome but also for its localization to the ~100 autosomal genes " (p5) in constrast to the statement 2 lines prior: "A similar number of Xist peaks across autosomes in ΔRepB cells was observed and the autosomal targets remained similar". Some quantitative statistics would assist in determining impact, both on autosomes and also X; perhaps similar to the quintile analysis done for expression.

We have added the Xist coverage panel for day 4 and 7 in the identified Xist-autosomal peak regions (new Figure 1E-F, Figure 2E-F), as mentioned above. The results clearly demonstrate that the deletion of RepB decreases Xist binding to autosomes. Also, we showed that ΔRepB increased X-linked genes expression in our revised Figure 3D.

It is stated that there is a significant suppression of X-linked genes with the autosomal transgenes; however, only an example is shown in Figure 4B. To support this statement, a full X chromosomal geneset should be shown in panels F and G, which should also list the number of replicates.

Please see new Figure 4B.

As these are hybrid cells, perhaps allelic suppression could be monitored? Is Med14 usually subject to X inactivation in the Ctrl cells, and is the expression reduced from both X chromosomes or preferentially the active (or inactive) X chromosome?

If Rev2 is referring to Figure 4, the dataset used in Figure 4 comes from another research group and was previously published (Loda, A. et al. Nat Commun, 2017).

If Rev2 is referring to our ES cells, they are N2 cell lines. The X chromosomes are fully hybridized (Cas/Mus), but the autosomes are not fully hybridized (Ogawa et al., Science, 2008). *Med14* is subject to XCI and is expressed from the Xa, silenced on the Xi.

The expression change for autosomes after transgene induction is barely significant; and it was not clear what was used as the Ctrl? This is a critical comparator as doxycycline alone can change expression patterns.

We agree that there was a modest change in expression after transgene induction, but it is a significant change. Again, the dataset is from a published study where the authors generated doxycycline-responsive Xist transgenes (see above). The control in this case is Dox-treated wildtype cells. We now clarify these points.

In the discussion there is the statement. "Genetic analysis coupled to transcriptomic analysis showed that Xist down-regulates the target autosomal genes without silencing them. This effect leads to clear sex difference - where female cells express the ~100 or so autosomal genes at a lower level than male cells (Figure 7H)." This sweeping statement fails to include that in MEFs there is no significant expression difference, in transgenics only borderline significance, and at d14 no significant expression difference. The down-regulation overall seems to be transient during development while targeting is ongoing?

Indeed, the Xist effects on autosomes seem to occur during cell differentiation in ES cells. While there is no apparent effect in MEFs, we cannot exclude effects on other somatic cells. Regardless of whether the effects are in early development or throughout life, the sex differences may have life-long effects in mammals. The study conducted in human cells by the Plath lab also concluded that the differences primarily affect stem cells.

Finally, I would have liked to see discussion of the consistency of the identified genes to support the conclusion that the autosomal sites are not merely the results of Xist diffusion.

We address this in the third paragraph of the Discussion. Our main argument is that if autosomal binding were caused by diffusion, then RepB deletion or X1 treatment would have led to increased binding at autosomal sites, as Xist would bind less to the X chromosome. However, as demonstrated in our study, both treatments resulted in reduced Xist binding on both the X chromosome and autosomes. This finding suggests that the binding is specific and reliant on Xist's RepA and RepB domains, rather than being a passive diffusion process.

To examine overlap between the conditions (days of differentiation and WT/RepB cells), we generated Venn Diagrams as now shown in Figure S4E.

The impact of Xist on autosomes is important for consideration of impact of changes in Xist expression with disease (notably cancers). Knowing the targets (if consistent) would enable assessment of such impact.

We thank Rev2 for the very helpful review and for the forward-looking experiments. Indeed, the physiological changes brought on by autosomal targeting will be of future interest.

**Reviewer #3 (Public review):**
Summary:Yao et al use CHART to identify chromatin associated with Xist in female mouse ESCs, and, as control, male ESCs at various timepoints of differentiation. Besides binding of Xist to X chromosome regions they found significant binding to autosomes, concentrating mostly on promoter regions of around 100 autosomal genes, as elucidated by MACS. The authors went on to show that the RepB repeat is mostly responsible for these autosomal interactions using a female ESC line in which RepB is deleted. Evidence is provided that Xist interacts with active autosomal genes containing lower coverage of repressive marks H3K27me3 and H2AK119ub and that RepB dependent Xist binding leads to dampening of expression, but not silencing of autosomal genes. These results were confirmed by overexpression studies using transgenic ESCs with doxycycline-inducible Xist as well as via a small molecule inhibitor of Xist (X1), inducing/inhibiting the dampening of autosomal genes, respectively. Finally, using MEFs and Xist mutants RepB or RepE the authors provide evidence that Xist is bound to autosomal genes in cells after the XCI process but appears not to affect gene expression. The data presented appear generally clear and consistent and indicate some differences between human and mouse autosomal regulation by Xist. Thus, these results are timely and should be published.

We thank Rev3 for the positive remarks and great suggestions. We have amended the manuscript per below.

Strengths:Regulation of autosomal gene expression by Xist is a "big deal" as misregulation of this lncRNA causes developmental defects and human disease. Moreover, this finding may explain sexspecific developmental differences between the sexes. The results in this manuscript identify specific mouse autosomal genes bound by Xist and decipher critical Xist regions that mediate this binding and gene dampening. The methods used in this study are appropriate, and the overall data presented appear convincing and are consistent, indicating some differences between human and mouse autosomal regulation by Xist.Weaknesses:(1) The figure legends and/or descriptions of data are often very short lacking detail, and this unnecessarily impedes the reading of the manuscript, in particular the figures would benefit not only from more detailed descriptions/explanations of what has been done but also what is shown.

We have included more detailed descriptions in the figure legends and throughout the manuscript.

This will facilitate the reading and overall comprehension by the reader. One out of many examples: In Fig S1B in the CHART data at d4 and d7 there is not only signal in female WT Xist antisense but also in female sense control. For a reader that is not an expert in XCI it would be helpful to point out in the legend that this signal corresponds to the lncRNA Tsix (I suppose), that is transcribed on the other strand.

We thank the reviewer for this excellent point. We have amended the Results section accordingly.

(2) Different scales are used in the lower panels of Figures 1A and 2A, which makes it difficult to directly compare signals between the different differentiation stages.

We have included a figure combining all timepoints — d0, d4, d7, and d14 WT female Xist CHART signals — on the X chromosome and autosomes to support our thesis. Please see new Figure 1B.

(3) In this study some of the findings on mouse cells contrast previously published results in human ESCs: (1) Xist binding occurs preferentially to promoters in mice, not in human. (2) Binding of Xist is mostly detected in polycomb-depleted regions in mice but there is a positive correlation between Xist RNA and PRC2 marks in human ESCs. These differences are surprising but may be very interesting and relevant. While I am aware that this might be a difficult task, it would be helpful to experimentally address this issue in order to distinguish whether species specific and/or methodological differences between the studies are responsible for these differences.

Indeed, our findings in mouse cells contrast with those observed in humans. As discussed in the manuscript, this discrepancy may be attributed to factors such as cell type, differentiation methods, and the Xist pull-down technique employed (our CHART method utilizes a 20 nt oligo library, whereas RAP uses long oligos). We agree that future work should investigate the underlying causes of these differences between mouse and human systems.

**Recommendations for the authors**:
**Reviewer #2 (Recommendations for the authors):**
For Figure 2: labelling ∆B on the panel A timeline (e.g. d0-∆B) would make the results clearer for the audience. Panel B makes most sense beside panel E of Figure 1, so combine here and skip in Figure 1?

We have modified Figure 2A and thank Rev2 for this suggestion. As for the embedded tables: since we performed peak calling for WT and ∆B separately, we believe that showing both the peak numbers and their corresponding peak patterns provides a clearer representation of the data.

I agree that at day 7 there appears to be a difference in X; but by day 14 this looks much more minimal - is it just time-shifted rather than altered? Perhaps this could be discussed. Autosomal binding sites show no change in number.

Day 7 exhibits the strongest Xist binding on the X chromosome, consistent with the de novo establishment phase of XCI when Xist is expressed at the highest levels 300 copies/cell during de novo XCI versus ~100 copies/cell during maintenance [Sunwoo et al., 2015 as cited]. Per our RNA-seq analysis here, we also observed highest Xist expression on day 7 and reduced levels on day 14 (Fig. S5A). This expression difference explains the reduced Xist CHART levels on day 14 compared to day 7.

While the X has previously been examined, it would seem beneficial to conduct the same expression analyses (Figure 3) for the X (perhaps supplemental), as the authors have the data 'in hand'. I feel comparison to X in the main figure for panels A and B would fit, while a similar analysis for the X for panel C could be supplemental, presumably supporting the published data to which this data is currently compared.

This is a good suggestion. Please find the new data in Figures 2E-F and 3D, which demonstrate that the RepB deletion inhibits Xist binding on the X chromosome, resulting in increased X-linked gene expression, as previously mentioned. Since Xist binds across the X chromosome, we did not perform peak calling as we did for the autosomes. Therefore, applying a similar analysis as in Figures 3A-B may not be appropriate in this case.

Such a direct comparison to X-data from the same study would be important. For panel H: How many replicates (2)? This should be in the legend. What is the change in median expression? Again, a supplemental figure showing impact on X-linked targets would be useful. Do male and female ESCs show an expression difference prior to differentiation (ie d0)? The data underlying this Figure should be in one of the supplementary tables, showing the full statistical tests and average change. The supplementary tables 8-12 list the WT target genes, not expression differences with the deletion. Again, given that the difference appears transient, might the ∆B cells be altered in rate of differentiation?

Panel H (revised Figure 3G) includes two replicates, and this has been added to the legends. We have provided a supplementary figure demonstrating that RepB increases the expression levels of X-linked genes on days 4, 7, and 14 (revised Figure 3D). Male and female ESCs show differences in the expression of X-linked genes, as both X chromosomes are active in females at this stage prior to differentiation (revised Figure S5C).

A supplementary table with statistical tests and average change information has been included in our revised version (Table S11).

On the other hand, these Xist-autosomal target genes displayed no significant differences between WT male, female, or ∆B female cells on day 0 — prior to onset of XCI and Xist expression. Please see new Figure 3H.

As for whether ∆B cells are altered in their rate of differentiation, the analysis by Colognori et al. 2019 indicates that ∆B cells differentiate similarly to WT cells. (In Figure 6 of Colognori et al. 2019, autosomal genes expressed similarly in WT and ∆B cells, whereas XCI is affected only in ∆B cells)

We have also modified the legends for our supplementary tables.

Why were the transgene lines examined upon neuronal differentiation rather than the same approach as in Figures 1-3? I would have thought neuronal differentiation might be more similar to d14, where limited changes remain? Could the authors clarify and discuss?

We apologize for the confusion. The Tg lines in Figure 4 came from a previously published study. We performed reanalysis of published datasets because we wanted to test whether — in the hands of other investigators — cell lines expressing Xist also supported autosomal targeting. Here we examined Tg1 and Tg2, which respond to doxycycline to overexpress Xist from an ectopic site. Transcriptomic analysis showed significant downregulation of autosomal Xist targets, as exemplified by *Bcl7b* and *Rbm14* (Figure 4C and S9B). In contrast, non-targets of Xist such as *Stau1* did not demonstrate significant changes in gene expression (Figure 4E and 4F). Looking across all autosomal target genes, we observed a significant decrease in mean expression in the Xist overexpressing cell lines (Figure 4D). The fact that the autosomal changes were also observed in datasets generated by other investigators greatly strengthen our conclusions. We have clarified this in the Results section.

Figure 5 - the legend should specify the number of replicates and clarify the blue/green (intuitive, but not specified). Are the 'target' / 'non-target' genes from d4 Chart (but the RNA from d5)? How are 'non-targets' defined - do they match the 'targets' in certain criteria (expression level, chromatin features, GC content)? Do they change per differentiation protocol?

We have modified the legends to clarify that the 'target' and 'non-target' genes are derived from the day 4 CHART-seq data, while the RNA data is from day 5, as that study sequenced day 5 and not day 4. Non-targets were randomly chosen based on (i) the absence of Xist binding and (ii) similar expression levels. Please see revised Figure S8.

It would be helpful to compare Xist expression levels across the various models, and the MEF model could be better described - are they polyploid as often happens?

We have included the Xist expression levels of ES cells and MEF cells in the revised version (revised Figure S5A, 6D). The transformed MEFs are indeed tetraploid, as is typical.

For 6A to be informative, one needs to know % mapping to X in ES timeline, which is in supplemental, so perhaps 6A should also be supplemental?

We have moved 6A to the supplemental figure.

It is odd that ∆B seems to have had more impact in MEFs, and I would like more discussion - but I also think I am missing something: "We observed that Xist signals were more substantially reduced on both the Xi and autosomal regions in ΔRepE MEFs compared to ΔRepB cells", yet in lower panel 6 G it looks like ∆B is LOWER than ∆E? Am I misinterpreting?

We apologize for the confusing writing. The revised text now reads: “To investigate, we utilized a deletion of Xist’s Repeat E (∆RepE), which was previously demonstrated to severely abrogate localization of Xist to the Xi 41,42. We reasoned that the severe loss of Xist binding might unmask a transcriptomic difference. As expected, we observed that Xist signals were somewhat more reduced on the Xi in ΔRepE MEFs compared to ΔRepB cells (Figure 6E-6F). Despite this reduction, peak coverages in autosomal target genes did not increase in ΔRepE MEFs (Figure 6E-6F). However, there was an overall decrease in the number of significant autosomal peaks in ∆RepE MEFs relative to WT cells (Figure 6A). Regardless, we observed no significant transcriptomic differences in ∆RepE MEFs relative to WT MEFs (Figure 7A-7E). Additionally, further examination of RNA sequencing data from male and female MEF cells in two published studies 43,44 corroborated that the expression levels of these autosomal Xist targets did not exhibit significant changes (Figure 7F and 7G). Altogether, the analysis in MEFs demonstrates that Xist continues to bind autosomal genes in post-XCI somatic cells. However, autosomal binding of Xist in post-XCI cells does not overtly impact expression of the associated autosomal genes. Nonetheless, we cannot exclude more subtle changes that do not meet the significance cut-off.”

Overall, I would like to see how consistent these autosomal peaks are - I shudder to suggest Venn diagrams, but something to show whether there are day/lineage specific peaks and/or ∆repeat B/E resistant peaks.

We now present Venn diagrams comparing MEF, ES_d4, and ES_d7, showing approximately 50% overlap between MEF and ES cells (revised Figure S10B). This may be expected, as each timepoint is a different developmental stage of XCI, with expected gene expression differences.

Very minor comments:It would be easier if the supplemental tables were tabs in 1 file!

We will defer to the editor on how best to format the supplemental tables.

Similar to the text, could gene names be included in the supplemental?

We have provided gene names in the supplemental files.

Figure 3 legend: should 'representing' be representative?

We have modified it.

"Xist patterns identified in human cells" p 5; it is challenging to follow human versus mouse, so specify or ensure correct use of XIST/Xist Indeed, we edited the manuscript accordingly.

Gene names should be italicized.

We have italicized gene names in our manuscript.

Ref. 38 lacks details (...).

We have updated the reference.

Peak-like characters - perhaps characteristics? P8

We have modified this.

**Reviewer #3 (Recommendations for the authors):**
On page 6, the 6th sentence in the first paragraph needs correction. "Consistent with Xist's behavior on the X chromosome."

We have modified the sentence. Thank you.